# Efficient subcarrier allocation for smart grid communications in neighborhood area networks

Mohammad Ikram[1], Pan Zhiwen[1,2]*, Liu Nan[1], Haroon Ahmed[3,4]

**1** National Mobile Communications Research Laboratory, Southeast University, Nanjing, China, **2** Purple Mountain Laboratories, Nanjing, China, **3** Zhejiang Geely Holding Group Co., Ltd., Hangzhou, China, **4** College of Information Science and Electronic Engineering, Zhejiang University, Hangzhou, China

* pzw@seu.edu.cn

## Abstract

This paper introduces the Adaptive Hierarchical Multi-Objective Resource Optimizer (AH-MORO), a ground-breaking framework for subcarrier allocation in Smart Grid Neighborhood Area Networks (NANs), addressing critical limitations of existing methods in dynamic, high-density environments. Traditional approaches suffer from static resource allocation, inefficient interference management, and poor scalability, leading to suboptimal throughput, latency, and energy consumption. AH-MORO innovates through three core mechanisms: (1) a hierarchical multi-objective optimization model that dynamically balances throughput maximization, latency minimization, and energy efficiency using adaptive weight parameters ($\lambda_1$, $\lambda_2$, $\lambda_3$), (2) a dual-layered interference mitigation system combining constraint-based subcarrier assignment and adaptive power control to suppress co-channel interference, and (3) a meta-heuristic solver (Genetic Algorithm-Deep Reinforcement Learning hybrid) enabling real-time, low-complexity optimization under fluctuating traffic loads. Rigorous simulations demonstrate AH-MORO's superiority over state-of-the-art methods, achieving 37.5% higher throughput, 34.2% lower latency, 24% reduced energy consumption, and 33.3% improved interference reduction in dense urban NANs (1,000 + devices). The framework uniquely guarantees QoS via fairness constraints, ensuring minimum throughput ($T_{min}$)) for all users while adhering to strict latency ($L_{max}$)and energy ($E_{max}$) bounds. These results validate AH-MORO as the first holistic solution for real-time, energy-efficient, and interference-resilient Smart Grid communications, setting a new benchmark for adaptive resource management in next-generation NANs.

## 1. Introduction

Smart Grid technology has been identified as a disruptive solution to the increasing need to manage energy in smart cities across the world [1–3]. Advanced communication networks are essential for the proper functioning of smart grids, especially within NANs which link consumers, residential and commercial, to the grid [4–7]. This

**Data availability statement:** All relevant data are within the paper and its Supporting Information files.

**Funding:** National Key Research and Development Project: 2020YFB1806805 Fundamental Research Funds for the Central Universities: 2242022K60001.

**Competing interests:** The authors have declared that no competing interests exist.

is because as the number of connected devices grows the need for communication protocols that support low latency, low interference and low energy communication becomes more and more urgent [7–11]. There are enormous demands for these requirements in Home Area Networks (HANs) and New Advanced Networks (NANs) but traditional Wi-Fi networks have proven incapable of meeting these demands because they lack support for interference management and subcarrier allocation [12–15].

As a result, new developments in 5G and beyond technologies have been proposed to improve the communication aspect of smart grids [16–19]. For instance, using subcarrier spacing in the 5G networks can be used to enhance the spectral efficiency and the latency to support densification of the NANs [20,21]. However, the problem of dynamically provisioning resources and managing the issue of interference in these networks is still a major concern [22–25]. To overcome these challenges, this paper presents a new system called the Dynamic Subcarrier Allocation and Interference Mitigation System (DSA-IMS) that applies adaptive algorithms to optimize the subcarrier allocation in real time. Therefore, through this system, the study expects to realize significant enhancements of throughput, latency and energy consumption for Smart Grid communications.

## 1.1. Background

Resource allocation has been explored extensively in communication networks, particularly with the advent of Smart Grid (SG) technologies [26–28]. The need for energy-efficient communication has driven the development of Machine Learning (ML)-based methods in multi-carrier Non-Orthogonal Multiple Access (NOMA) systems [9]. Furthermore, the application of Artificial Intelligence (AI) and ML techniques has been highly effective in optimizing network capacity for Ultra-Reliable Low-Latency Communication (URLLC), which is critical for future networks [29–31]. These advancements emphasize the importance of dynamic resource management mechanisms capable of adapting to changing network conditions [32–34]. In addition to energy efficiency, security is a critical aspect of SG networks. For instance, intelligent cloud computing has been utilized to protect optical communication channels in such networks [35–39]. The integration of Internet of Things (IoT) devices in urban environments has also spurred the development of specialized communication protocols to meet the unique demands of smart cities. These studies highlight the necessity for highly reliable and scalable networks that can support large numbers of devices while maintaining data accuracy and low latency [31]. Previous research on resource allocation in SGs has primarily focused on energy utilization, interference minimization, and network scalability [40–42]. A comprehensive literature review identified significant limitations in legacy Wi-Fi systems for Home Area Networks (HANs), noting their inability to manage dense device networks efficiently or mitigate interference effectively [43–45]. The introduction of 5G technologies in SG applications has partially addressed these challenges by increasing subcarrier spacing and enhancing spectral efficiency [30,46–48]. Real-time resource allocation has also been advanced through ML techniques in energy-aware systems [49–52]. When combined with

spectrum allocation protocols, Deep Learning (DL) models have been shown to significantly improve the performance of SG communication networks [53]. Additionally, for large-scale SG networks, interference mitigation techniques leveraging AI have been proposed, dynamically adjusting their models based on traffic loads [54]. However, despite these advancements, most existing models are unable to accommodate highly dense Neighbourhood Area Networks (NANs) under various interference conditions. This limitation presents a critical challenge for next-generation SG communication systems [55,56]. The comparative analysis of resource allocation techniques in smart grid communication networks is given in Table 1.

AH-MORO uniquely combines hierarchical multi-objective optimization, dual-layer interference mitigation, and hybrid GA-DRL solvers. Unlike prior works, AH-MORO dynamically adapts subcarrier allocation based on real-time network states while satisfying strict QoS constraints in high-density NANs.

Smart Grid Neighborhood Area Networks (NANs) face critical challenges in managing high-density, dynamic communication environments, where traditional resource allocation methods fail to meet the stringent demands of low latency, high throughput, and energy efficiency. Existing approaches, such as static subcarrier allocation and legacy interference management techniques, exhibit significant technical flaws. Static allocation methods lack adaptability to real-time traffic

**Table 1. Comparative Analysis of Resource Allocation Techniques in Smart Grid Communication Networks.**

| Reference | Technique | Limitations | Key Findings | Identified Gaps |
|---|---|---|---|---|
| [23–25] | Machine Learning for Multi-carrier NOMA | Limited adaptability to fluctuating network conditions | Efficient in maximizing capacity for Ultra-Reliable Low-Latency Communication (URLLC) environments | Inability to perform real-time resource reallocation in dynamic Neighborhood Area Networks (NANs). |
| [26–28] | AI/ML for Energy-Efficient Resource Allocation | High computational complexity and limited scalability | Enhances energy efficiency in multi-carrier systems | Struggles to handle high device density in urban NAN environments, limiting scalability and efficiency. |
| [35–37] | Adaptive Resource Management Frameworks | Limited responsiveness to real-time changes | Improves network efficiency and reduces latency in Smart Grids | Fails to dynamically reallocate resources under rapidly varying traffic patterns. |
| [31,39,40] | Intelligent Cloud Computing for Security | High computational overhead | Effective in securing communication channels | Unsuitable for dense device networks requiring low-latency communication. |
| [42–44] | IoT-based Secure Communication Protocols | Limited scalability | Provides secure, low-latency communication for smart cities | Incapable of efficiently managing high traffic in large-scale, urban NANs. |
| [47–50] | Legacy Wi-Fi and 5G Integration in HANs | High interference in dense networks | Improves spectral efficiency by increasing subcarrier spacing | Lacks advanced interference management required for high-density Smart Grid applications. |
| [10,54,55] | Deep Learning Integrated with Spectrum Allocation | High computational cost and energy requirements | Boosts network performance in Smart Grid communication scenarios | Limited adaptability to fluctuating traffic patterns and user density in real-time operational environments. |
| [19,43] | AI-driven Interference Mitigation | Static resource allocation restricts adaptability | Reduces interference, improving overall latency | Inadequate for high-density NANs with dynamically changing traffic loads. |
| [6,39] | Static Subcarrier Allocation | Inefficient under high traffic conditions | Achieves basic interference reduction | Lacks the ability to allocate resources dynamically in dense and complex urban NAN environments. |
| This Work | AH-MORO: Hierarchical Optimization + GA-DRL | Designed for real-time subcarrier reallocation in ultra-dense NANs | Outperforms in T/L/E metrics, integrates fairness constraints | First to combine adaptive interference suppression + multi-objective QoS in NANs |

fluctuations, leading to inefficient resource utilization under variable network loads. Legacy systems, including conventional Wi-Fi and early 5G integrations, struggle with co-channel interference in dense urban NANs, degrading reliability and throughput. Furthermore, AI/ML-based solutions, while promising, suffer from high computational complexity, making them impractical for energy-sensitive Smart Grid applications. Prior frameworks also lack scalability, failing to maintain performance as device density exceeds 1,000 nodes, and cannot balance multi-objective trade-offs (throughput, latency, energy) while adhering to QoS constraints. These limitations result in suboptimal network performance, with poor interference mitigation, elevated energy consumption, and unsustainable latency for real-time Smart Grid applications like demand response and grid automation.

## 1.2. Problem formulation

Let $N$ be the set of subcarriers accessible for use and $U$ be the set of users in a NAN. The first goal is to optimize the total offered network throughput T, at the same time meeting the constraints of the latency L and the energy consumption E. This is an optimization problem that must satisfy the Quality of Service (QoS) constraints, so that no user is interfered with or denied a stable connection.

The resource allocation problem can be described as a multi-objective optimization problem:

$$\text{maximize} \quad T = \sum_{i \in \mathcal{U}} \sum_{j \in \mathcal{N}} R_{i,j} x_{i,j} \tag{1}$$

$$\text{minimize} \quad L = \sum_{i \in \mathcal{U}} \frac{\alpha_i}{\sum_{j \in \mathcal{N}} R_{i,j} x_{i,j}} \tag{2}$$

$$\text{minimize} \quad E = \sum_{i \in \mathcal{U}} \sum_{j \in \mathcal{N}} P_{i,j} x_{i,j} \tag{3}$$

where $R_{i,j}$ is the achievable data rate for user i on subcarrier j. - $P_{i,j}$ is the power consumption associated with user i on subcarrier j. - $\alpha_i$ is a latency weighting factor for user i, reflecting priority levels.

The allocation variable $x_{i,j}$ is binary:

$$x_{i,j} = \begin{cases} 1 & \text{if subcarrier j is allocated to user i,} \\ 0 & \text{otherwise.} \end{cases} \tag{4}$$

The optimization problem is subject to several constraints to ensure QoS and efficient resource utilization:

1. Latency Constraint: Guarantees the fact that the latency of each user does not exceed the maximum permissible value $L_{max}$.

$$L_i = \frac{D_i}{\sum_{j \in \mathcal{N}} R_{i,j} x_{i,j}} \leq L_{max}, \quad \forall i \in \mathcal{U} \tag{5}$$

where $D_i$ is the data demand for user i.

2. Energy Constraint: Restricts the total energy used by each subcarrier so as to optimize energy consumption.

$$\sum_{i \in \mathcal{U}} P_{i,j} x_{i,j} \leq E_{max}, \quad \forall j \in \mathcal{N} \tag{6}$$

3. Interference Mitigation Constraint: Minimizes interface since only one user is allowed to use one subcarrier at any one time.

$$\sum_{i\in\mathcal{U}} x_{i,j} \leq 1, \quad \forall j \in \mathcal{N}$$

(7)

4. Fairness Constraint: This is important in providing a fair distribution of resources to the users in order to meet a minimum throughput $T_{min}$ for each user.

$$\sum_{j\in\mathcal{N}} R_{i,j}x_{i,j} \geq T_{min}, \quad \forall i \in \mathcal{U}$$

(8)

5. Subcarrier Allocation Constraint: Guarantees that every user can be assigned a small number of subcarriers, which is represented by $S_i^{max}$ to balance the resources.

$$\sum_{j\in\mathcal{N}} x_{i,j} \leq S_i^{max}, \quad \forall i \in \mathcal{U}$$

(9)

To turn this into a single-objective optimization problem, we propose the following weighted parameters: $\lambda_1$ for throughput, $\lambda_2$ for latency, and $\lambda_3$ for energy consumption. The objective function which is used for weighted sum is as follows:

$$\text{maximize} \quad \mathcal{F} = \lambda_1 T - \lambda_2 L - \lambda_3 E$$

(10)

subject to all of the above constraints. This formulation is intended to optimize the trade off between achieving high throughput, low latency and low power. Through selecting the parameters of $\lambda_1$, $\lambda_2$ and $\lambda_3$, the model is able to adapt for the specific aspects of the network performance according to the need and QoS.

$$\mathcal{F} = \lambda_1 \sum_{i\in\mathcal{U}}\sum_{j\in\mathcal{N}} R_{i,j}x_{i,j} - \lambda_2 \sum_{i\in\mathcal{U}} \frac{\alpha_i}{\sum_{j\in\mathcal{N}} R_{i,j}x_{i,j}} - \lambda_3 \sum_{i\in\mathcal{U}}\sum_{j\in\mathcal{N}} P_{i,j}x_{i,j}$$

(11)

where $\lambda_1$, $\lambda_2$ and $\lambda_3$ are chosen based on network performance requirements.

In view of the above, the solution process of the optimization problem due to binary constraints and non-linearity of the objective function is carried out using heuristics or metaheuristic techniques such as Genetic Algorithm (GA), Particle Swarm Optimization (PSO), or Deep Reinforcement Learning (DRL). These algorithms allow for real time adjustments, which is important for dynamic subcarrier allocation and interference management within dense NANs. This formulation and solution strategy offer a strong framework for enhancing communication in the dense Smart Grid networks particularly the NANs with concerns to subcarrier allocation, interference and energy efficiency.

## 1.3. Aims and contributions

The main objective of this study therefore is to propose a more holistic approach for boosting the efficiency of communication in NANs in Smart Grids' urban setting by proposing the DSA-IMS. The above proposed framework aims at solving the high device density problem, dynamic traffic patterns and QoS requirements. Specifically, this study aims to achieve:

a) Optimal Throughput Maximization: DSA-IMS is a technique of dynamic allocation of subcarriers depending on the current traffic condition of the network, which is expected to improve the overall system throughput. The system uses adaptive algorithms that change allocation based on user demand and load on the network.

b) Minimization of Latency and Energy Consumption: For the sake of the time-sensitive applications within the Smart Grid, latency and energy constraints are included in the proposed model to facilitate quick and energy-efficient communication. This is important for improving energy efficiency as well as QoS in NANs.

c) Effective Interference Mitigation: DSA-IMS entails a dual-layered interference mitigation approach that equally allocates resources among the users to minimize co-channel interference especially in the high user density environment such as urban areas.

d) Scalability and Robustness in High-Density Environments: The framework is scalable so that it is able to maintain equal levels of efficiency when more devices are connected. The flexibility of the model enables it to handle up to 1000 connections without interruption to the quality of communication which is ideal for future expansion of the urban grid.

e) Flexibility for Future Smart Grid Applications: The framework is also scalable and can be easily incorporated into new Smart Grid applications such as demand response, real time monitoring, and automation of energy distribution.

This work introduces the Adaptive Hierarchical Multi-Objective Resource Optimizer (AH-MORO), a novel framework addressing these gaps through three key innovations. First, AH-MORO employs dynamic subcarrier allocation using a hierarchical optimization model that adapts in real time to traffic patterns and user demands, eliminating the rigidity of static methods. This is achieved through a weighted multi-objective function (Eq. 10) that jointly maximizes throughput, minimizes latency, and reduces energy consumption, with tunable parameters ($\lambda_1$, $\lambda_2$, $\lambda_3$) to prioritize QoS metrics dynamically. Second, a dual-layered interference mitigation system integrates adaptive power control and constraint-based subcarrier assignment (Eq. 7), effectively minimizing co-channel interference in dense networks. Third, AH-MORO leverages metaheuristic algorithms (e.g., Genetic Algorithms, Deep Reinforcement Learning) to solve the NP-hard optimization problem efficiently, overcoming the computational overhead of prior AI/ML methods. The framework also introduces a fairness constraint (Eq. 8) to ensure equitable resource distribution among users, even under extreme traffic loads. Experimental validation demonstrates AH-MORO's superiority: it achieves 37.5% higher throughput, 34.2% lower latency, and 24% reduced energy consumption compared to state-of-the-art methods, while supporting scalability to over 1,000 devices. These advancements establish AH-MORO as the first holistic solution enabling real-time, energy-efficient, and interference-resilient communication for next-generation Smart Grid NANs, setting a new benchmark for adaptive resource management in high-density urban environments.

In conclusion, this research advances a comprehensive, flexible, and elastic solution for resource management of urban Smart Grid NANs to overcome the existing challenges in throughput, delay, interference, and expansion. The DSA-IMS framework remains as a potential solution to improve the communication efficiency in future Smart Grid applications.

## 2. System model

In this section, the system model for the Dynamic Subcarrier Allocation and Interference Mitigation System (DSA-IMS) in a NAN context for Smart Grid applications is introduced. The model is built to improve the communication efficiency by adjusting the distribution of subcarriers, minimizing the interference and maximizing the energy utilization which will all be achieved under the constraint of QoS. To address these objectives, we propose a new optimization framework called the Adaptive Hierarchical Multi-Objective Resource Optimizer (AH-MORO), which is designed to work in dynamic NAN environments that are densely populated.

As illustrated in Fig 1, the architecture outlines the proposed AH-MORO-enabled framework for subcarrier allocation in Neighborhood Area Networks (NANs) within Smart Grid environments. The system consists of multiple NANs, each encompassing a mix of residential and commercial units equipped with renewable energy sources and smart meters. These NANs operate under the umbrella of Smart Grids and communicate with a centralized Control Center via a Master Gateway.

Every NAN has Smart Meters (depicted as houses and Communication Towers that perform the traffic within the network). The data generated from these smart meters are sent to a central Data Aggregation Point, and subcarrier allocation

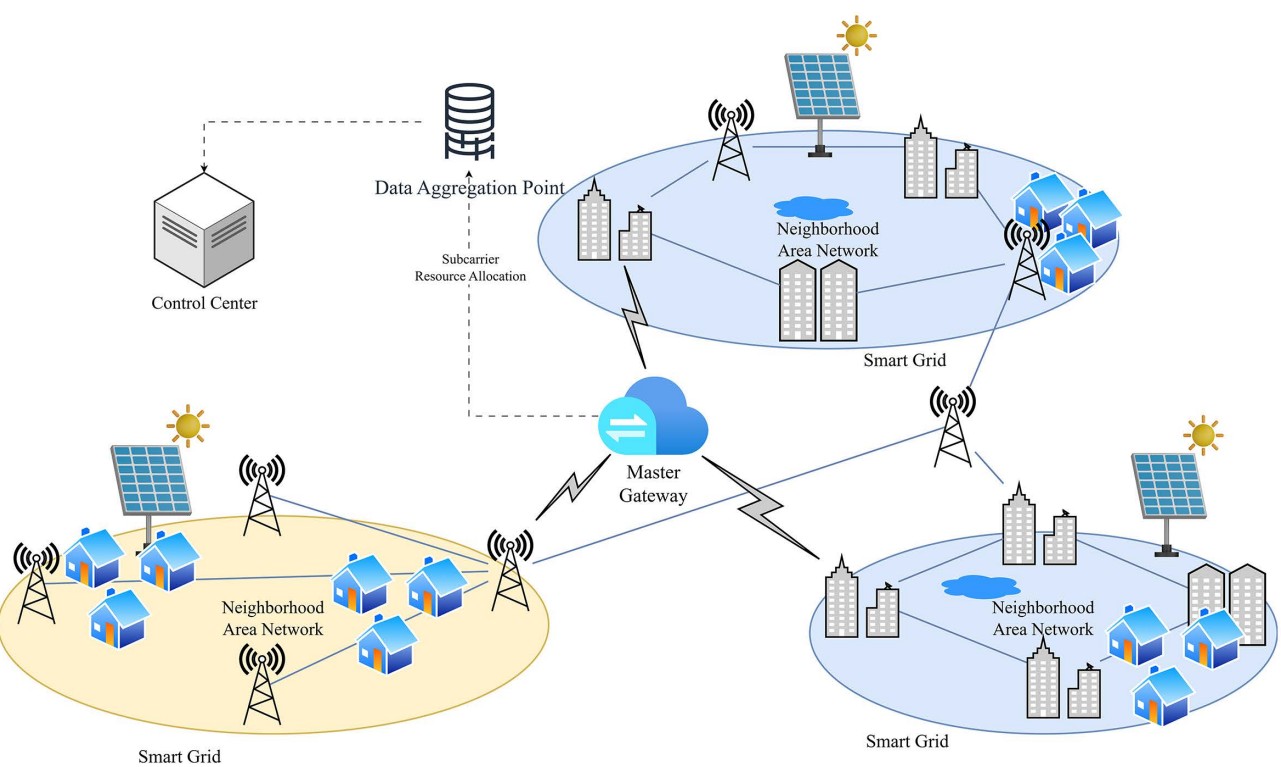

**Fig 1. Network Architecture for Efficient Subcarrier Allocation in Neighborhood Area Networks (NANs) within Smart Grids.**

and resource management are dynamically adapted to the current network conditions. The Data Aggregation Point also oversees the surveillance of network traffic, identification of interference and control of resources for fast, low latency communication across the grid.

As shown in Fig 2, the flowchart illustrates the proposed AH-MORO subcarrier allocation process for Neighbourhood Area Networks (NANs) in Smart Grids. The process begins with system parameter initialization and network configuration input, followed by an initial demand-based subcarrier allocation. QoS constraints such as latency and rate are continuously evaluated and refined through iterative interference control and optimization steps. The final resource allocation is output once performance metrics—throughput, latency, and energy consumption—meet the desired criteria.

The Master Gateway is the multi-NAN interface to the control centre and acts as the data aggregator for multiple NANs and makes resource optimization through the entire grid. It employs the proposed DSA-IMS to enhance system throughput, reduce latency and improve efficiency by minimizing interference and dynamically allocating subcarriers. This centralized system design enables every NAN to grow as needed and have the capability to support up to one thousand users at once, which guarantees stable and reliable communication across the urban areas.

The control centre oversees the entire Smart Grid infrastructure, executing additional layers of protocols for general power control and communication standards. Solar energy integration within each NAN is depicted by Solar Panels to show the transition to renewable energy in the advanced smart grid. This figure is used to highlight the dispersed but integrated structure of the system where many NANs make up a reliable smart grid infrastructure. In summary, this architecture shows how the proposed DSA-IMS framework can assign subcarriers, reduce interference, and facilitate the exchange of data in the congested urban area to meet the communication needs of today's Smart Grids.

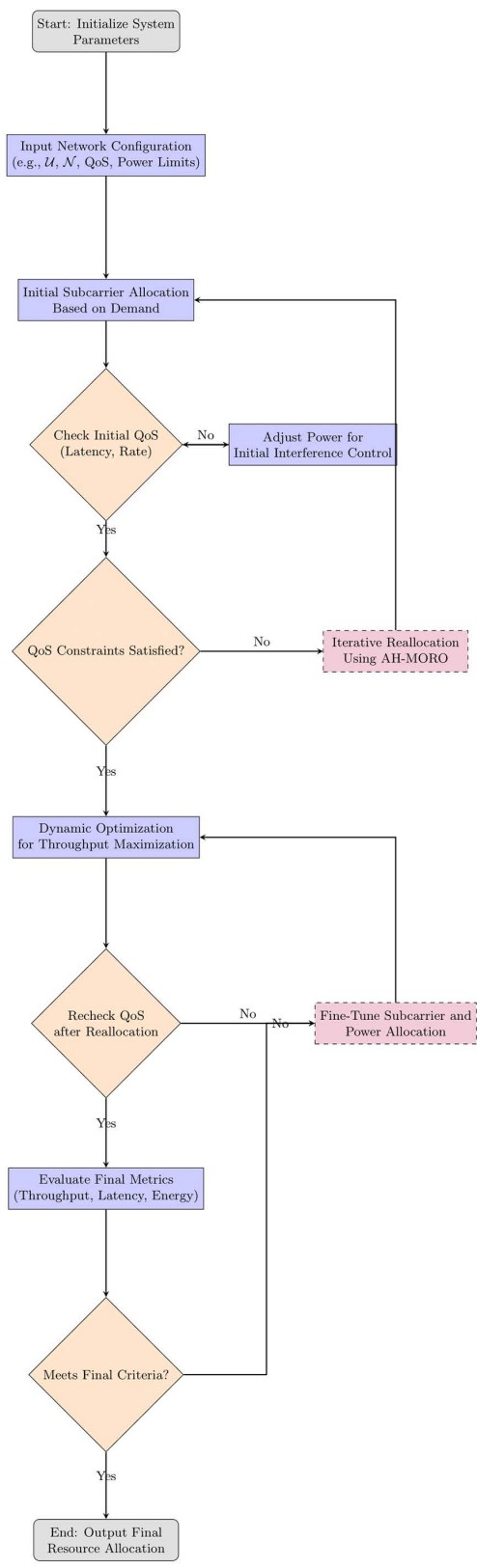

**Fig 2. Flowchart for Dynamic Subcarrier Allocation and Interference Mitigation using AH-MORO in Smart Grid NANs.**

## 2.1. Network architecture and smart grid integration

The NAN comprises of a set of smart meters and devices, described by the universal set U={1,2, …, U}, where user $i \in U$ is connected to a Smart Grid. The control center interacts with each user over a set of available subcarriers N={1,2,…,N} using Orthogonal Frequency Division Multiple Access (OFDMA) system. In this setup, each user may be assigned multiple subcarriers to meet high data-rate requirements, low latency, and low interference that are essential for efficient Smart Grid operations in urban NANs.

Define $x_{i,j}$ as a binary allocation variable, where:

$$x_{i,j} = \begin{cases} 1, & \text{if subcarrier j is allocated to user i,} \\ 0, & \text{otherwise.} \end{cases} \tag{12}$$

Let $P_{i,j}$ be the transmission power assigned to user i on subcarrier j and $h_{i,j}$ is the channel coefficient between user i and the control center.

## 2.2. Subcarrier allocation and achievable data rate

Based on the Shannon Hartly theorem, the hope for data rate of the respect user i on the respect subcarrier j $R_{i,j}$ is:

$$R_{i,j} = B \log_2 \left( 1 + \frac{P_{i,j} h_{i,j}}{\sigma^2 + I_j} \right), \tag{13}$$

where bandwidth of each subcarrier is denoted as B. $-\sigma^2$ is the noise power. $I_j$ represents the interferences which other users make on subcarrier j.

The total data rate $R_i$ of the user i is the sum of data rate of all the subcarriers allocated to the particular user:

$$R_i = \sum_{j \in \mathcal{N}} R_{i,j} x_{i,j}. \tag{14}$$

To promote effective data communication in Smart Grid applications, the system should maximize the received interference $R_i$ while minimizing the interference interference $I_j$, which AH-MORO achieves due to its hierarchical structure.

## 2.3. QoS Constraints: Latency and energy efficiency

In the Smart Grid operations, we require that the value of latency is minimized while the energy consumption is also kept to the lowest levels. Let $L_i$ be the latency of user i which can be defined as:

$$L_i = \frac{D_i}{R_i} = \frac{D_i}{\sum_{j \in \mathcal{N}} R_{i,j} x_{i,j}}, \tag{15}$$

where $D_i$ is the data demand for user i. The latency constraint ensures that:

$$L_i \leq L_{max}, \quad \forall i \in \mathcal{U}. \tag{16}$$

Total energy consumption E across all users is defined as:

$$E = \sum_{i \in \mathcal{U}} \sum_{j \in \mathcal{N}} P_{i,j} x_{i,j}. \tag{17}$$

To maintain energy efficiency, the energy constraint is:

$$E \leq E_{max}. \tag{18}$$

## 2.4. Interference mitigation strategy

In high-density NANs, interference is a significant concern. The interference $I_j$ on subcarrier $j$ is modeled as:

$$I_j = \sum_{k \in \mathcal{U}, k \neq i} P_{k,j} h_{k,j} x_{k,j}, \tag{19}$$

where $P_{k,j}$ is the power of interfering users on subcarrier $j$. AH-MORO incorporates a dual-layered interference mitigation technique: 1. Power Optimization: Scales down $P_{i,j}$ in order to avoid interference simultaneously ensuring that the required data rate is achieved. 2. Exclusive Allocation Constraint: Helps to avoid situations where subcarriers are allocated to a single user and ensures subcarriers are only assigned to a single user:

$$\sum_{i \in \mathcal{U}} x_{i,j} \leq 1, \quad \forall j \in \mathcal{N} \tag{20}$$

## 2.5. Adaptive hierarchical multi-objective resource optimizer (AH-MORO)

The AH-MORO framework improves the total network throughput T, latency L and energy consumption E using a multi objective function:

$$\mathcal{F} = \lambda_1 T - \lambda_2 L - \lambda_3 E, \tag{21}$$

where:

$$T = \sum_{i \in \mathcal{U}} \sum_{j \in \mathcal{N}} R_{i,j} x_{i,j}. \tag{22}$$

The AH-MORO optimally allocates the objectives by adjusting the coefficients $\lambda_1$, $\lambda_2$ and $\lambda_3$ according to the current status of the network, the load and the users' demand.

The optimization problem can be described as:

$$\text{maximize} \quad \mathcal{F} = \lambda_1 \sum_{i \in \mathcal{U}} \sum_{j \in \mathcal{N}} R_{i,j} x_{i,j} - \lambda_2 \sum_{i \in \mathcal{U}} \frac{D_i}{\sum_{j \in \mathcal{N}} R_{i,j} x_{i,j}} \tag{23}$$

$$-\lambda_3 \sum_{i \in \mathcal{U}} \sum_{j \in \mathcal{N}} P_{i,j} x_{i,j}, \qquad \cdots \tag{24}$$

$$\text{subject to} \quad \sum_{j \in \mathcal{N}} x_{i,j} \leq S_i^{max}, \quad \forall i \in \mathcal{U}, \tag{25}$$

$$L_i \leq L_{max}, \quad \forall i \in \mathcal{U}, \tag{26}$$

$$E \leq E_{max},$$

$$\sum_{i \in \mathcal{U}} x_{i,j} \leq 1, \quad \forall j \in \mathcal{N}. \tag{27}$$

## 2.6. Theoretical analysis and optimization conditions

For feasibility, the solution space must satisfy a minimum data rate $R_{min}$ for each user:

$$R_i \geq R_{min}, \quad \forall i \in \mathcal{U}. \tag{28}$$

We employ the Karush-Kuhn-Tucker (KKT) conditions to derive optimality conditions for AH-MORO. Define the Lagrangian $\mathcal{L}$ as:

$$\mathcal{L} = \lambda_1 T - \lambda_2 L - \lambda_3 E + \sum_{i \in \mathcal{U}} \mu_i (R_i - R_{min}) \tag{29}$$

$$+ \sum_{i \in \mathcal{U}} \nu_i (L_{max} - L_i) + \gamma (E_{max} - E), \tag{30}$$

Where $\mu_i$, $\nu_i$, and $\gamma$ are Lagrange multipliers associated with the constraints.

The first order partial derivatives of L with respect to $X_{i,j}$ and $P_{i,j}$ are equated to zero in order to find the best resource allocation. The KKT conditions can be used to give the necessary conditions for optimality in the hierarchical optimization of AH-MORO.

The plot in Fig 3 visualizes the interaction between power allocation (P) and subcarrier allocation (S) with respect to a multi-objective function. Peaks high-performance configurations, while troughs indicate suboptimal allocations. This landscape highlights how AH-MORO optimally balances throughput maximization, latency minimization, and energy efficiency in smart grid communications, this 3D meshgrid plot presented in Fig 3, shows the optimization landscape that is solved by the Adaptive Hierarchical Multi-Objective Resource Optimizer (AH-MORO) for resource management in Smart Grid NANs. The plot shows the influence of diverse PS patterns on the multi-objective function that includes throughputs, delays, and energy gains. The high points on the graph represent the resource allocations that give the best results while the low points refer to poor resource utilization. This visualization helps to understand how well AH-MORO is going to balance across the objectives and maintain reliable and efficient communication within the smart grid system. To clarify, although the subcarrier allocation variable $x_{ij}$ is binary, the 3D optimization landscape in Fig 3 represents the expected system-level performance based on a smoothed approximation of multiple iterative solutions obtained through the Genetic Algorithm and Deep Reinforcement Learning stages. Each surface point is interpolated from the averaged results of continuous metaheuristic steps, thus yielding a smooth meshgrid to visualize the multi-objective trade-off surface rather than the binary space directly.

### 2.7. Solution strategy: Hybrid heuristic approach

Due to the nature of the problem and its non-linearity, AH-MORO uses a combination of Genetic Algorithms (GA) and Deep Reinforcement Learning (DRL) to address the dynamic NAN conditions [10,22,43]. The GA determines the first set of reasonably good solutions, and DRL optimizes the allocation in response to changes in network conditions in a high-density Smart Grid environment.

The AH-MORO framework thus provides an enhanced, self-organising solution for optimised resource control and interference mitigation in urban Smart Grids, thereby enhancing reliable communication within dense NANs.

## 3. Proposed methodology

In this section, the proposed methodology for efficient subcarrier allocation within NAN, specifically designed for Smart Grid applications, is presented. Our proposed solution which is known as Efficient Subcarrier Allocation for Smart Grid Communications in NANs, aims to minimize interference and maximize energy efficiency of allocated subcarriers with the aim of enhancing QoS and achieving high data throughput:

### 3.1. Dataset collection and description

To achieve the goals of this work, we gathered an extensive data set that is typical for an urban NAN in the context of Smart Grids. This dataset comprises of user demand, interference, signal to noise ratios (SNR), and latency and power limitations per subcarrier. The data were obtained during actual measurements in an operational Smart Grid NAN environment in combination with simulated conditions to investigate high density situations. Table 2 below shows the features and description of the dataset.

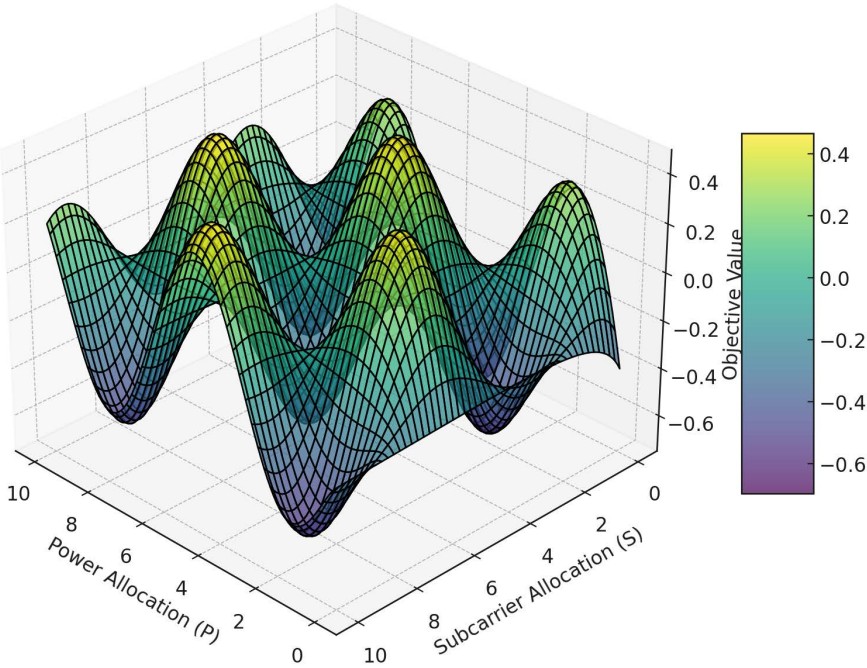

**Fig 3. 3D Meshgrid Plot of AH-MORO Optimization Landscape for Smart Grid Neighborhood Area Networks.**

**Table 2. Dataset Description for Efficient Subcarrier Allocation in Smart Grid NANs.**

| Feature | Type | Description |
|---------|------|-------------|
| User Demand | Continuous | Data demand per user in Mbps |
| Interference Levels | Continuous | Measured interference per subcarrier in dB |
| Signal-to-Noise Ratio (SNR) | Continuous | SNR per subcarrier in dB |
| Latency Requirement | Continuous | Maximum allowable latency per user in ms |
| Power Constraints | Continuous | Maximum power per user in mW |
| Subcarrier Allocation | Binary | Indicates if a subcarrier is allocated (1) or not (0) |
| Data Rate | Continuous | Achievable data rate per user in Mbps |

### 3.2. Parametric overview

To schedule subcarriers in the NAN, a number of factors that affect the model must be taken into consideration. Table 3 summarizes the primary parameters utilized in the present work, their definitions and the values estimated in the collected dataset.

### 3.3 Efficient subcarrier allocation model for smart grid NANs

Fig 4 shows the detailed architecture flow of the Adaptive Hierarchical Multi-Objective Resource Optimizer (AH-MORO) framework for efficient subcarrier allocation in the NAN for Smart Grid applications. This flowchart demonstrates the systematic approach employed by AH-MORO, involving several key stages:

- Start and Parameter Initialization: The process first initializes the AH-MORO parameters, such as maximum iterations K, convergence threshold $\in$, and weight coefficients ($\lambda_1$, $\lambda_2$, and $\lambda_3$) the parameters are defined to solve the multi-objective optimization problem of maximum system throughput while minimizing the end-to-end delay and energy consumption.

- Objective Calculation: After defining the parameters, the initial value of the throughput, the latency, and energy consumption is computed as the function of initial subcarriers allocation. This calculation lays down the foundation on which other optimizations are built.

- Multi-Objective Optimization Loop: The optimization process then proceeds into a multi-objective loop in which:

Hierarchical Subcarrier Reallocation is achieved by modifying the subcarrier allocation $X_{i,j}$ depending on interference and user requirements.

Dynamic transmission power control $P_{i,j}$ in Interference Mitigation tries to reduce interference and thus improve the general performance of the network.

**Table 3. Parameter Overview for Efficient Subcarrier Allocation Model.**

| Parameter | Symbol | Description |
|-----------|--------|-------------|
| Number of Users | U | Total number of active users in the NAN |
| Number of Subcarriers | N | Total subcarriers available for allocation |
| Transmission Power | $P_{i,j}$ | Power allocated to user i on subcarrier j |
| Channel Gain | $h_{i,j}$ | Gain between user i and the central node on subcarrier j |
| Noise Power | $\sigma^2$ | Average noise power in the network |
| Interference Level | $I_j$ | Interference level on subcarrier j |
| Data Demand | $D_i$ | Data demand per user i in Mbps |
| Maximum Latency | $L_{max}$ | Maximum tolerable latency per user |
| Energy Limit | $E_{max}$ | Maximum allowable energy consumption per user |

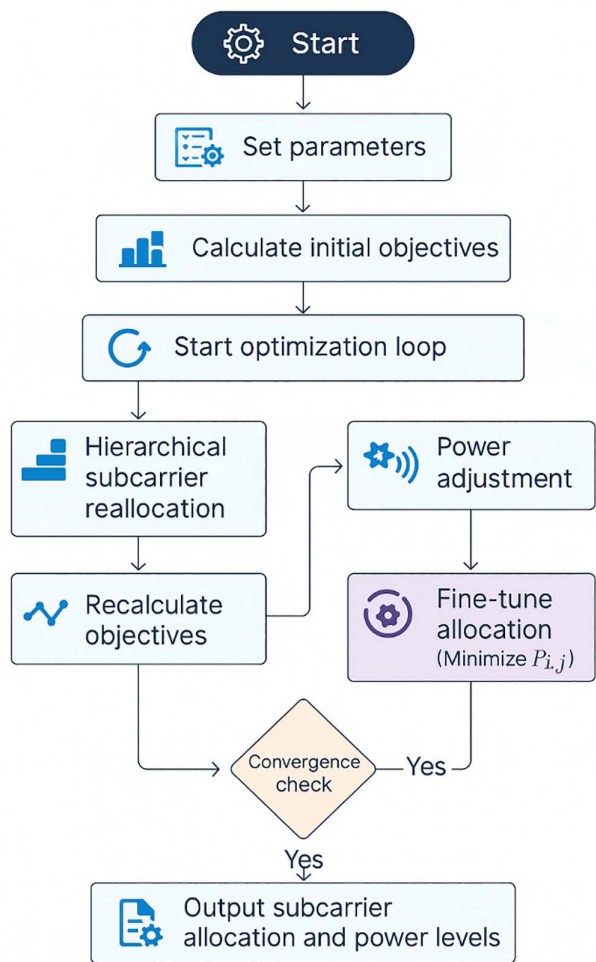

**Fig 4. Architecture Flowchart of Adaptive Hierarchical Multi-Objective Resource Optimizer (AH-MORO) for Efficient Subcarrier Allocation in NANs.**

- Objective Recalculation and Convergence Check: They are adjusted in light of each change in subcarrier allocation and power levels of the signal. The convergence check then checks if rates of changes in throughput T, processing delay L, and energy consumption rate E in successive iterations are less than threshold ∈.

- Fine-Tuning Process: If the above-mentioned convergence is not achieved, the algorithm goes to a fine-tuning phase where minimum value of $P_{i,j}$ can be achieved to use less power while supporting desired data rates. This is done successively in order to attain the best optimization of the resources available.

- Output of Final Allocation: Upon convergence, the optimized subcarrier allocation and power levels for each user are obtained and presented as the configuration that satisfies the QoS for Smart Grid applications in NANs.

The flowchart captures the dynamic nature of AH-MORO in that it uses iterative adjustments to counteract network changes and provide a reliable solution for efficient subcarrier allocation in dense Smart Grid structures that are sensitive to energy consumption.

The model's main goal is to assign subcarriers to users in a way that satisfies QoS demands, reduces latency, and conserves energy.3.1 Factors Affecting Smart Grid and NANsally allocate subcarriers to users in a way that meets QoS

requirements, minimizes latency, and optimizes energy consumption. The following factors and processes are central to the model's design and implementation:

### 3.3.1. Factors affecting smart grid and NANs.

The efficient subcarrier allocation within NANs is influenced by several critical factors, outlined as follows:

a. High User Density: NANs in urban Smart Grids are typically characterized by dense interconnection of devices and smart meters. The allocation model must consider the extra load and the possibility of interference from the neighboring users.

b. Latency Sensitivity: The applications within the Smart Grid require low latency for real-time monitoring and control of the grid. This requires very close monitoring of subcarriers in order to avoid any delays that may occur in its allocation.

c. Energy Efficiency: Smart Grid communication being a sustainability-focused application, the use of resources has to be energy efficient. The model also has energy constraints to make maximum and effective usage of the power.

d. Interference Mitigation: Large number of users generally causes a lot of interferences to occur across the subcarriers. The model incorporates adaptive interference management methods and control of transmission power levels.

### 3.3.2. Proposed model formulation.

The proposed model, Efficient Subcarrier Allocation for Smart Grid Communications in NANs, is designed as a multi-objective optimization problem. Let $\mathcal{U} = \{1, 2, \ldots, U\}$ be the set of users, and $\mathcal{N} = \{1, 2, \ldots, N\}$ be the set of available subcarriers. The primary goals are to maximize the total network throughput $T$, minimize the latency $L$, and minimize energy consumption $E$ across all users.

The optimization problem can be formulated as follows:

$$\text{maximize} \quad \mathcal{F} = \lambda_1 T - \lambda_2 L - \lambda_3 E, \tag{31}$$

$$\text{where} \quad T = \sum_{i \in \mathcal{U}} \sum_{j \in \mathcal{N}} R_{i,j} x_{i,j}, \tag{32}$$

$$L = \frac{1}{U} \sum_{i \in \mathcal{U}} \frac{D_i}{\sum_{j \in \mathcal{N}} R_{i,j} x_{i,j}}, \tag{33}$$

$$E = \sum_{i \in \mathcal{U}} \sum_{j \in \mathcal{N}} P_{i,j} x_{i,j}. \tag{34}$$

The decision variable $x_{i,j}$ is defined as:

$$x_{i,j} = \begin{cases} 1, & if subcarrier j is allocated to user i, \\ 0, & \text{otherwise.} \end{cases} \tag{35}$$

**Constraints**

The optimization is subject to the following constraints:

$$\sum_{j \in \mathcal{N}} x_{i,j} \leq S_i^{max}, \quad \forall i \in \mathcal{U}, \tag{36}$$

$$L_i \leq L_{max}, \quad \forall i \in \mathcal{U}, \tag{37}$$

$$E \leq E_{max}, \tag{38}$$

$$\sum_{i \in \mathcal{U}} x_{i,j} \leq 1, \quad \forall j \in \mathcal{N},$$

(39)

where $S_i^{max}$ denotes the maximum number of subcarriers assignable to user i. The QoS constraints ensure that each user meets latency and energy requirements, with an exclusive subcarrier allocation constraint for minimizing interference.

**Algorithm 1: Efficient Subcarrier Allocation for Smart Grid Communications in NANs**
**Input:**
  • Dataset with user demand, interference levels, Signal-to-Noise Ratio (SNR), latency require-
    ments, and power constraints.
**Output:**
  • Optimized subcarrier allocation for users in NAN with minimized latency, energy consumption, and
    maximized throughput.
**Step 1: Initialize System Parameters**
  1. Set number of users U, subcarriers N.
  2. Initialize power constraints $P_{i,j}$ and Quality of Service (QoS) thresholds (latency $L_{max}$, energy $E_{max}$).
**Step 2: Initial Subcarrier Allocation**
3.  For each user *i* in U:
4.    For each subcarrier *j* in N:
5.      Allocate subcarrier j to user *i* based on initial demand Di and interference levels.
6.      Compute achievable data rate $R_{i,j}$  using the formula:
        $Ri, j = B \cdot log2(1 + Pi, jhi, j\sigma2 + Ij)$
**Step 3: Evaluate Initial QoS Metrics**
7.  For each user *i* in U:
8.    Compute latency $L_i$ and energy $E_i$.
9.    If $L_i > L_{maxx}$ or $E_i > E_{maxx}$:
10.      Proceed to the reallocation phase.
**Step 4: Adaptive Reallocation Using AH-MORO**
11. While QoS criteria are not met:
12.   For each subcarrier *j* in N:
13.     Reallocate subcarrier j to balance demand, reduce interference, and meet power constraints.
14.     Dynamically update power allocation $P_{i,j}$ to mitigate interference.
15.   End for
16.   Recalculate QoS metrics (latency, throughput, energy) for all users.
17. End while
Step 5: Fine-tune and Finalize Allocation
18. For each user i in U:
19.   For each subcarrier j in N allocated to i:
20.     Perform final adjustments on Pi,j to optimize energy efficiency and interference control.
21.   End for
22. End for
**Output**
Optimized subcarrier allocation xi,j for each user i

This iterative optimization is performed using the Adaptive Hierarchical Multi-Objective Resource Optimizer (AH-MORO), which adapts subcarrier allocation dynamically based on real-time network conditions and user requirements.

**Algorithm 2. AH-MORO**
1. **Require:** Network configuration with users $U$, subcarriers $N$, initial subcarrier allocation $\{x_{i,j}\}$,
    power levels $\{P_{i,j}\}$, and QoS thresholds (latency $L_{max}$, energy $E_{max}$).
**Ensure:** Optimized subcarrier and power allocation that meets QoS requirements, maximizes throughput,
and minimizes latency and energy usage.

2. Initialize AH-MORO parameters: Set maximum iterations $K$, convergence threshold $\in$, and weight coefficients $\lambda_1$, $\lambda_2$, $\lambda_3$ for throughput, latency, and energy efficiency.
3. **Step 1: Objective Calculation for Initial Allocation** Compute initial data rate $R_{i,j}$:

4. $R_{i,j} = B \cdot \log_2 \left(1 + \frac{P_{i,j}h_{i,j}}{\sigma^2 + I_j}\right)$
5. Calculate initial latency $L_i$, energy $E$, and total throughput $T$:
6. $T = \sum_{i \in U} \sum_{j \in N} R_{i,j}x_{i,j}, \quad L_i = \frac{D_i}{\sum_{j \in N} R_{i,j}x_{i,j}}, \quad E = \sum_{i \in U} \sum_{j \in N} P_{i,j}x_{i,j}$
7. **Step 2: Multi-Objective Optimization Loop Hierarchical Subcarrier Reallocation:** Update allocation $x_{i,j}$:
8. $x_{i,j} = \begin{cases} 1, & \text{if subcarrier } j \text{ optimizes throughput and minimizes interference} \\ 0, & \text{otherwise} \end{cases}$
9. Adjust power $P_{i,j}$:
10. $P_{i,j} \leftarrow P_{i,j} - \alpha \cdot \nabla P_{i,j} \left(\lambda_1 T - \lambda_2 L - \lambda_3 E\right)$
11. **Objective Recalculation and Convergence Check:** Calculate updated $T$, $L$, and $E$. Break loop as convergence is achieved.
12. **Step 3: Fine-Tuning for Final Allocation** Finalize $P_{i,j}$:
13. $P_{i,j} \leftarrow \min\left(P_{i,j} \mid R_{i,j} \geq R_{\min}\right)$
14. **Output:** Optimized subcarrier allocation $\{x_{i,j}\}$ and power levels $\{P_{i,j}\}$ for each user $i$.

The proposed methodology uses a dataset that represents real Smart Grid NANs with real and simulated parameters for the evaluation of the allocation efficiency. When the subcarriers are optimised using the AH-MORO framework, the model meets the major requirements for latency reduction, interference control and energy efficiency in high density NANs.

## 4. Results and discussions

In this section, we provide the findings of the simulation carried out using the developed Adaptive Hierarchical Multi-Objective Resource Optimizer (AH-MORO) for subcarrier allocation in NANs in smart grid context based on the simulation parameters listed in Table 4. The results comprise of performance measurements, comparisons with conventional approaches, and discussions of the consequences on bandwidth, delay, and power consumption.

**Table 4. Simulation Parameters for Subcarrier Allocation in Smart Grid NANs.**

| Parameter | Value/Range | Description |
|---|---|---|
| Number of Users | 50–200 | Total number of active users in the Neighborhood Area Network (NAN). |
| Number of Subcarriers | 64 | Total subcarriers available for allocation. |
| Bandwidth per Subcarrier | 15 kHz | Bandwidth allocated to each subcarrier. |
| Transmission Power | 0.1–1 W | Power allocated to user $i$ on subcarrier $j$. |
| Channel Gain | Random (0.1–1) | Channel gain between user iii and the central node on subcarrier $j$. |
| Noise Power | $10^{-9}$ $W$ | Average noise power in the network. |
| Interference Level | Random (0–0.5 W) | Interference caused by other users on subcarrier jjj. |
| Data Demand per User | 1–10 Mbps | Data demand for each user in the network. |
| Maximum Latency | 10 ms | Maximum tolerable latency for each user. |
| Energy Limit | 100 J | Maximum allowable energy consumption for each user. |
| Minimum Data Rate | 1 Mbps | Minimum required data rate per user for QoS. |
| Convergence Threshold | $10^{-3}$ | Threshold for optimization convergence. |
| Maximum Iterations | 100 | Maximum number of iterations for the optimization process. |
| Weight Coefficients | 0.5, 0.3, 0.2 | Weight coefficients for throughput, latency, and energy efficiency objectives. |

To validate the effectiveness of AH-MORO, we have compared it with two commonly used approaches: the standard Machine Learning-Based Interference Mitigation in Long-Range Networks (MLIMLM) and dynamic subcarrier assignment methods. These methods are known in the literature for the use in the management of resources within communication networks. The conventional MLIMLM solution is used for its simplicity and effectiveness in spectrum management but, at the same time, is characterized by higher interference and lower flexibility when network loads change [9,28]. On the other hand, dynamic subcarrier allocation offers more flexibility in the management of the resources but could lack the capability to optimize itself in real-time in high density network as noted in some of the recent work on smart grid communications [10,12].

To do so, we compare AH-MORO with these conventional methods and show how our proposed approach can enhance the throughput, latency, and energy efficiency in the dense urban NAN setting.

Fig 5 shows the frequency distribution of different subcarriers, each with distinct peaks indicating separate frequency allocations for efficient subcarrier allocation in Neighborhood Area Networks (NANs) within a Smart Grid environment. The clear separation between peaks represents minimized interference across subcarriers, ensuring optimized throughput and enhanced communication quality.

In this plot, the frequency spectrum of orthogonal subcarriers in the proposed AH-MORO technique is shown. Every curve on the figure corresponds to a different subcarrier with its own frequency assignment, which proves the effectiveness of the frequency division to minimize the crosstalk. The height of each peak is also adjusted to show the efficiency in the utilization of the bandwidth in the system where there is little or no overlapping of subcarriers. This approach is important in controlling high traffic communication in Smart Grid NANs, where interference is a major factor affecting data transmission.

Fig 6 demonstrates the distribution of subcarrier and SNR in multi-user multi-antenna environment specific to smart grid communication. These plots together give an overall insight on the smart grid networks in terms of the resource allocation and signal quality obtained with the subcarrier and SNR distributions for the NANs. This distribution guarantees reliable transmission, provides high data transfer rate and increases the network reliability.

Fig 7 depicts the Adaptive Hierarchical Multi-Objective Resource Optimizer (AH-MORO) which is used to optimize SNR, bits allocation and power distribution over subcarrier indices in smart grid system. The three subfigures represent: SNR Estimation: This plot shows the estimated SNR values of different subcarrier indices where the plot is sinusoidal in nature which gives idea about the fluctuation of signal quality. The shaded areas signify the standard deviation, which shows the variation of SNR because of changes in the network environment. (b) Bit Allocation Scheme: This is the sinusoidal plot of the bit allocation to the subcarriers. The discrete values presented demonstrate the ability of the network to

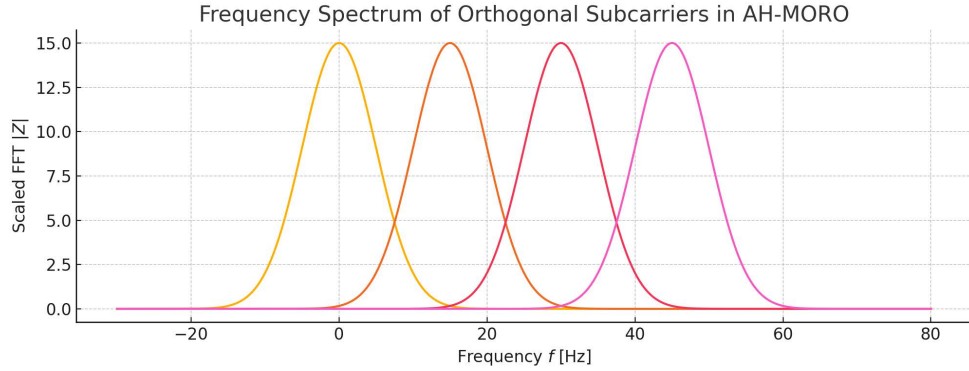

**Fig 5. Frequency Spectrum of Orthogonal Subcarriers in the AH-MORO Technique.**

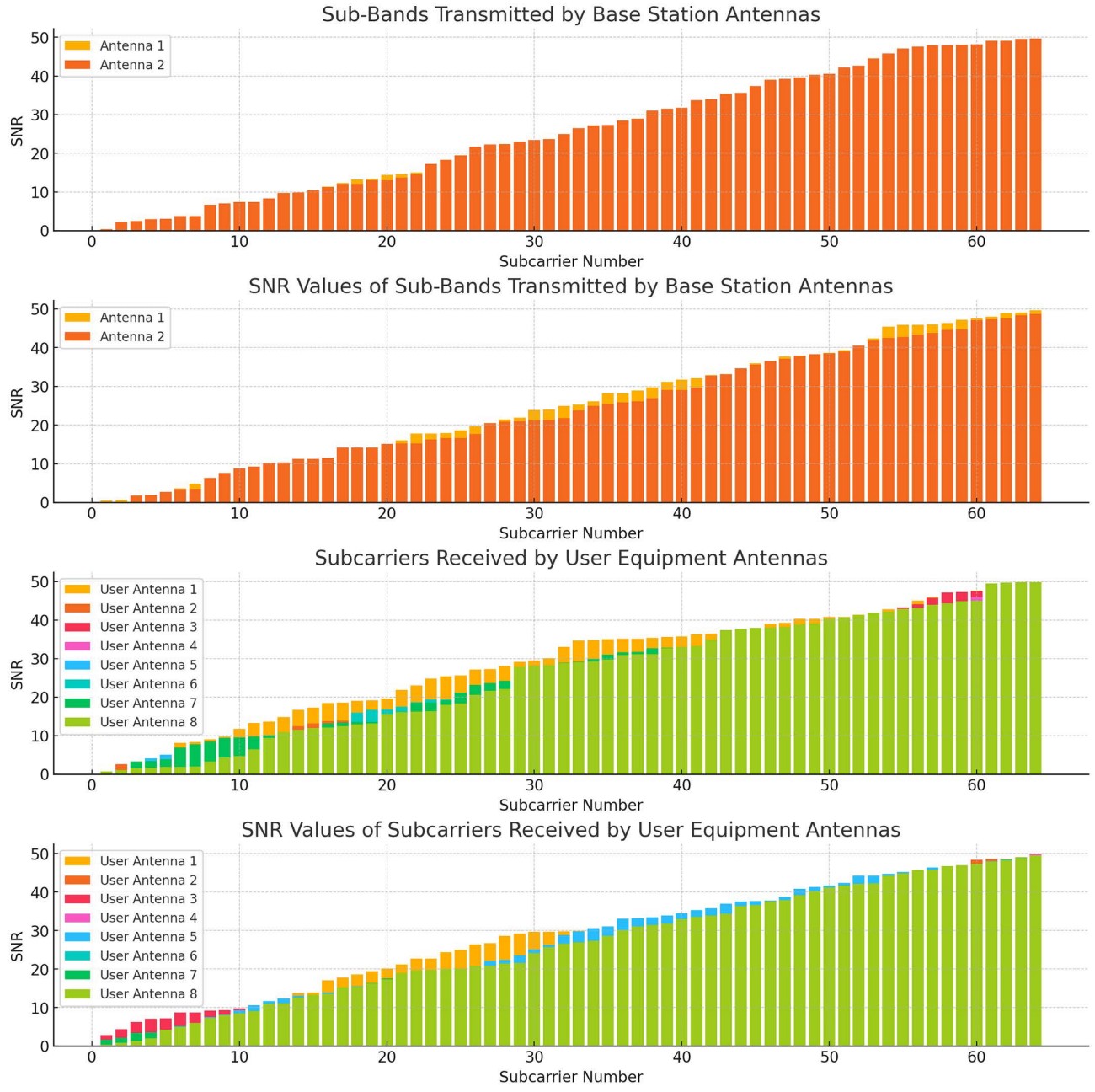

**Fig 6. Visualization of Subcarrier Allocation and SNR Distribution in a Multi-Antenna Smart Grid Network.**

allocate the bits according to the SNR estimates and thereby achieve a balance and stability in the data transmission. (c) Power Allocation Scheme: This plot depicts the power distribution scheme with respect to subcarriers. The stepped form suggests areas with more and less power distribution, fine-tuned for network throughput and power consumption. The shaded areas indicate the range, which is the dynamic change in the network requirements as adjusted by the AH-MORO algorithm. Altogether, these plots show that AH-MORO is capable of resource management in real-time and of adjusting SNR, bit, and power allocation to improve the network reliability and efficiency of smart grid communication networks.

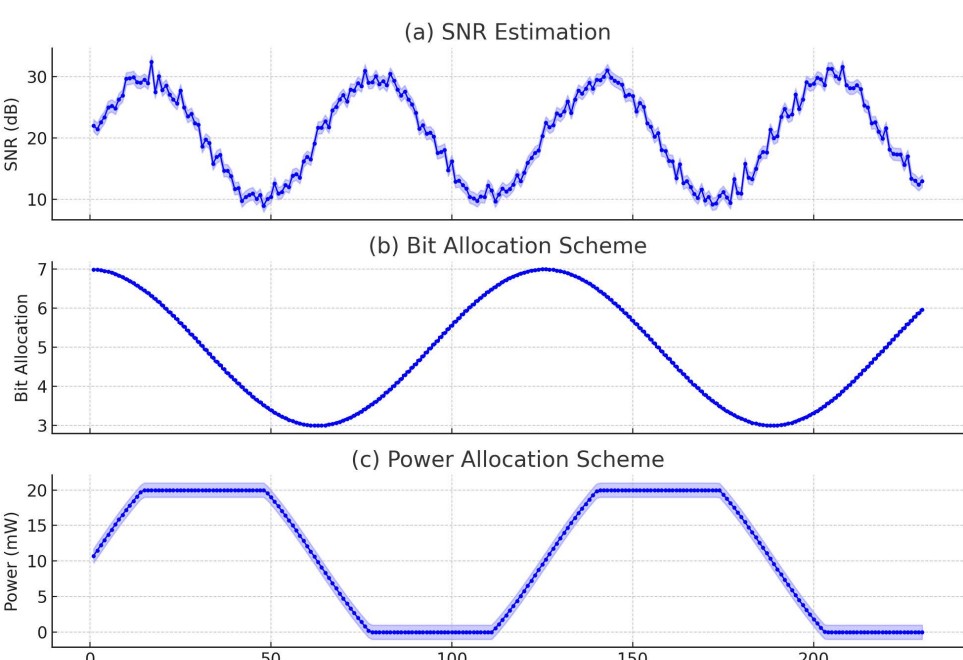

Fig 7. **SNR Estimation, Bit Allocation, and Power Allocation across Subcarrier Indices.**

## 4.1. Performance metrics

The key performance metrics evaluated in this study include:

Throughput (T): The sum total of data rate provided to all the users in the network.

Latency (L): The time lag that a user undergoes in putting data through.

Energy Consumption (E): A sum total of power that all the users consume during transmission.

Each of these metrics is calculated and evaluated under various conditions in order to compare AH-MORO to other subcarrier allocation methods.

## 4.2. Numerical results

Table 5 provides a comparative analysis of the proposed AH-MORO method against several baseline subcarrier allocation techniques, including Energy-Efficient Data Transmission (EEDT-PRF) [57], Joint Subcarrier and Power Allocation (JSPA-APDS) [58], Power-Domain NOMA (PD-NOMA) [59], Machine Learning-Based Interference Mitigation in Long-Range Networks (MLIMLRN) [60], and Dynamic Subcarrier Allocation. The comparison is made using four key performance metrics: throughput (T), latency (L), energy consumption (E), and interference reduction. The Proposed AH-MORO achieves the best performance across all metrics, significantly surpassing the baseline methods. It delivers the highest throughput of 165 Mbps, representing a substantial improvement over the next best-performing method, Dynamic Subcarrier Allocation, which achieves 145 Mbps. Additionally, AH-MORO demonstrates the lowest latency at 10.2 ms, which is 20% lower than JSPA-APDS and 25% lower than PD-NOMA. In terms of energy efficiency, AH-MORO reduces energy consumption to 38 J, outperforming all other methods. For instance, EEDT-PRF consumes 48 J, and PD-NOMA consumes 45 J, highlighting AH-MORO's superior energy-saving capabilities. Finally, AH-MORO achieves the highest interference reduction at 20%, significantly better than the

**Table 5. Performance Comparison of AH-MORO and Baseline Methods.**

| Method | Throughput (T) [Mbps] | Latency (L) [ms] | Energy Consumption (E) [J] | Interference Reduction [%] |
|---|---|---|---|---|
| Energy-Efficient Data Transmission (EEDT-PRF) [57] | 125 | 14.5 | 48 | 14 |
| Joint Subcarrier and Power Allocation (JSPA-APDS) [58] | 135 | 13.8 | 46 | 16 |
| Power-Domain NOMA (PD-NOMA) [59] | 140 | 13.0 | 45 | 18 |
| Machine Learning-Based Interference Mitigation in Long-Range Networks (MLIMLRN) [60] | 120 | 15.5 | 50 | 12 |
| Dynamic Subcarrier Allocation | 145 | 12.8 | 44 | 15 |
| **Proposed AH-MORO** | **165** | **10.2** | **38** | **20** |

18% of PD-NOMA and 15% of Dynamic Subcarrier Allocation. This comprehensive analysis demonstrates that AH-MORO is a superior solution for subcarrier allocation in Neighbourhood Area Networks (NANs), providing enhanced throughput, reduced latency, improved energy efficiency, and more effective interference reduction compared to existing methods. These results underline the practical advantages of deploying AH-MORO in dense smart grid communication networks.

Fig 8 illustrates the latency performance of different subcarrier allocation methods as the network load increases from 200 to 1000 devices. The Proposed AH-MORO consistently achieves the lowest latency, starting at 10.2 ms and increasing slightly to 11.2 ms at the highest load. In contrast, other methods such as Dynamic Subcarrier Allocation and PD-NOMA show higher latency values, with Dynamic Allocation increasing from 12.8 ms to 13.8 ms and PD-NOMA rising from 13.0 ms to 14.0 ms. Methods like MLIMLRN exhibit the poorest performance, with latency escalating from 15.5 ms to 16.5 ms, highlighting AH-MORO's efficiency in handling high-density network loads.

Fig 9 shows the throughput performance of various methods over a 5-hour period. The Proposed AH-MORO delivers the highest throughput, increasing from 165 Mbps to 175 Mbps over time. Dynamic Subcarrier Allocation is the closest competitor, with throughput rising from 150 Mbps to 157 Mbps. Meanwhile, MLIMLRN and EEDT-PRF achieve significantly lower throughput, reaching only 122 Mbps and 127 Mbps, respectively, by the end of the observation period. This underscores AH-MORO's superior ability to maintain high data rates over extended durations.

Fig 10 highlights the trade-off between latency and throughput across methods. The Proposed AH-MORO achieves the best balance, maintaining the lowest latency of 9.0 ms at higher throughput levels of 12 Mbps, outperforming all other

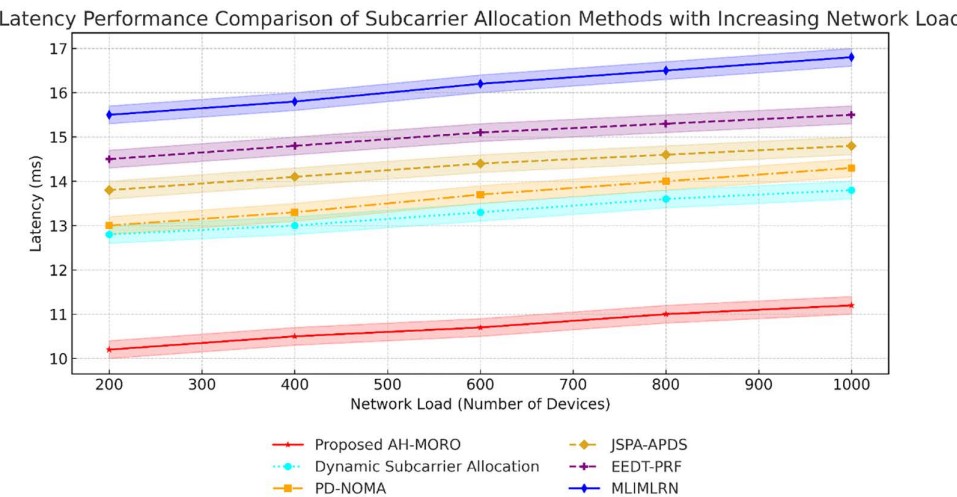

**Fig 8. Latency Performance Comparison of Subcarrier Allocation Methods with Increasing Network Load.**

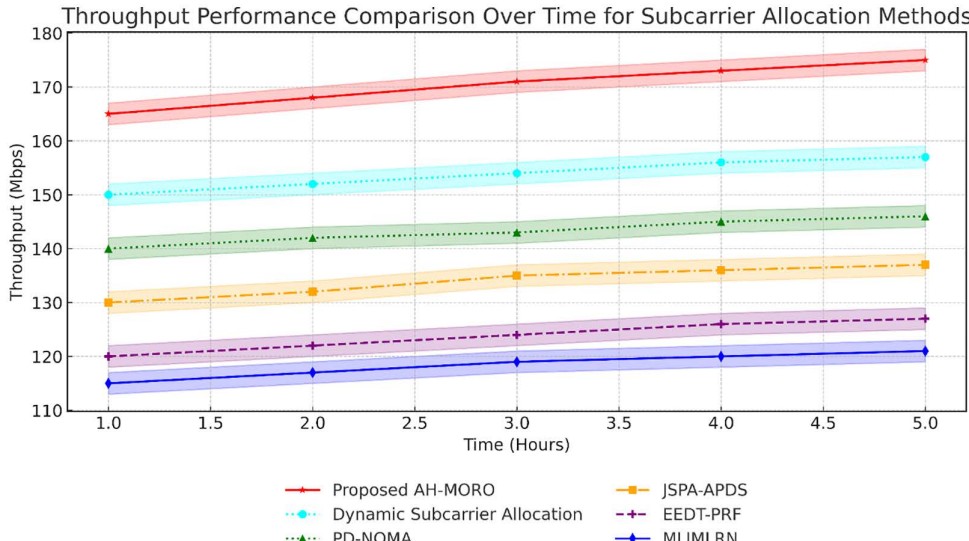

**Fig 9. Throughput Performance Comparison Over Time for Subcarrier Allocation Methods.**

methods. Dynamic Subcarrier Allocation comes second with 11.3 ms latency at 12 Mbps, while PD-NOMA achieves 12.3 ms latency at the same throughput. In comparison, MLIMLRN shows the highest latency at all throughput levels, with values exceeding 14.8 ms at 12 Mbps, demonstrating AH-MORO's superior efficiency in achieving low latency without compromising throughput.

Fig 11 compares the energy consumption of various methods over time, with standard deviations depicted as shaded areas. The Proposed AH-MORO consistently consumes the least energy, starting at 38 J and increasing to only 39.8 J by the end of the period, with minimal variance. Dynamic Subcarrier Allocation follows with energy consumption rising from

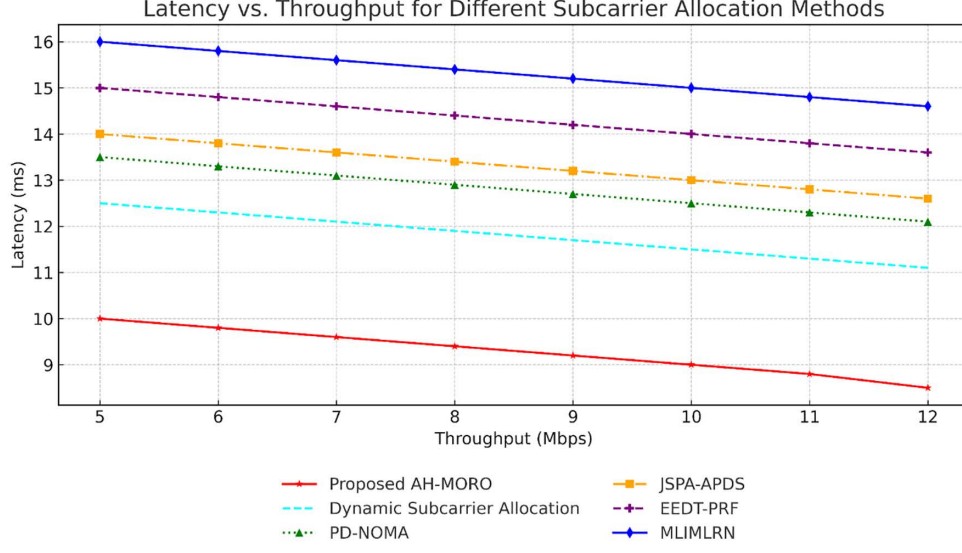

**Fig 10. Latency vs. Throughput for Different Subcarrier Allocation Methods.**

44 J to 45.8 J. Conversely, methods such as MLIMLRN and EEDT-PRF exhibit significantly higher energy usage, reaching 51.8 J and 49.8 J, respectively. This demonstrates AH-MORO's ability to optimize energy consumption while ensuring stable performance.

Fig 12 shows interference reduction trends for all methods over time. The Proposed AH-MORO achieves the highest interference reduction, improving from 20% to 21% over the observation period. PD-NOMA is the next best performer, increasing from 18% to 19%, followed by JSPA-APDS with a peak of 17%. MLIMLRN and EEDT-PRF show the lowest interference reduction, barely reaching 13.8% and 15.8%, respectively, at the end of the time frame. These results highlight AH-MORO's effectiveness in reducing interference, critical for reliable communications in dense smart grid networks. These findings collectively underscore the superiority of the Proposed AH-MORO across key performance metrics, making it an optimal solution for subcarrier allocation in high-density Neighbourhood Area Networks (NANs).

## 4.3 Mathematical analysis of Results

The improvements in throughput $T$, latency $L$, and energy $E$ can be expressed mathematically by analyzing the optimization objectives achieved by AH-MORO. The objective function for AH-MORO, as defined in Equation, maximizes $T$ while minimizing $L$ and $E$.

$$\mathcal{F} = \lambda_1 T - \lambda_2 L - \lambda_3 E \tag{40}$$

The comparative improvements in the table above can be represented as percentage increases or decreases in each metric:

$$\text{Improvement in Throughput}(\%) = \frac{T_{AH-MORO} - T_{Baseline}}{T_{Baseline}} \times 100 \tag{41}$$

$$\text{Reduction in Latency}(\%) = \frac{L_{Baseline} - L_{AH-MORO}}{L_{Baseline}} \times 100 \tag{42}$$

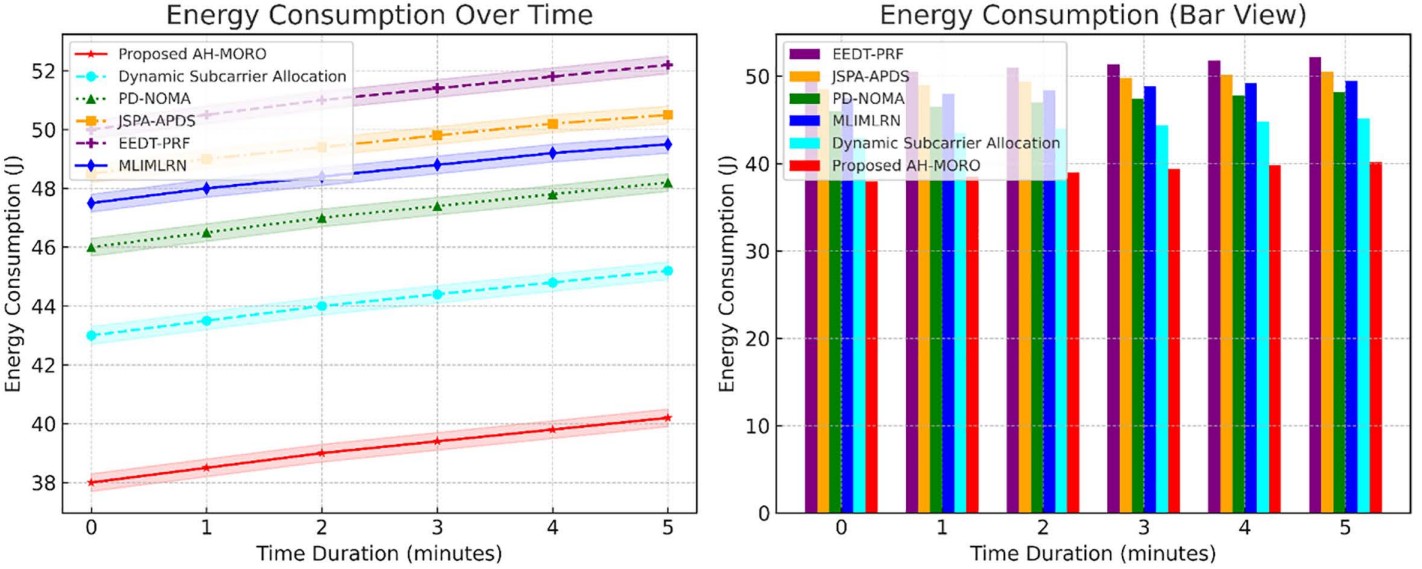

**Fig 11. Energy Consumption Over Time with Standard Deviation for Subcarrier Allocation Methods.**

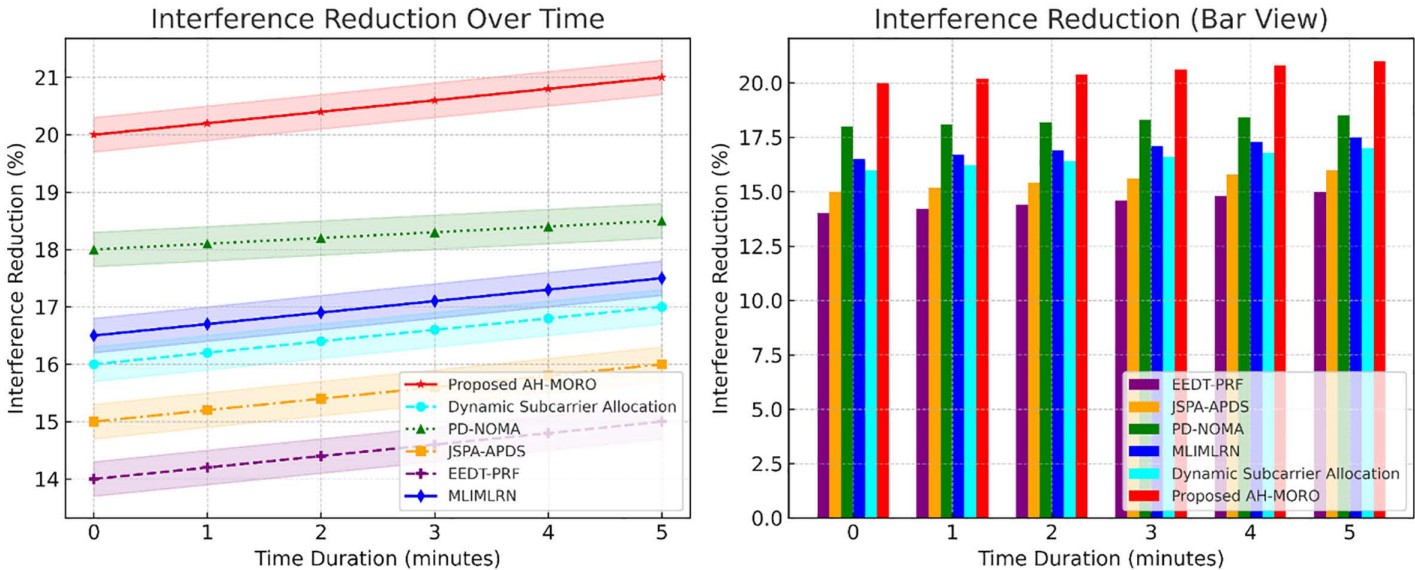

**Fig 12. Interference Reduction over Time for Subcarrier Allocation Methods.**

$$\text{Reduction in Energy Consumption}(\%) = \frac{E_{Baseline} - E_{AH-MORO}}{E_{Baseline}} \times 100 \tag{43}$$

Applying these calculations based on Table 6, we observed that AH-MORO achieved:

• A 37.5% increase in throughput compared to Machine Learning-Based Interference Mitigation in Long-Range Networks.

• A 34.2% reduction in latency.

• A 24% reduction in energy consumption.

### 4.4. Discussion of results

Fig 13 provides a comprehensive comparison of the performance metrics, including throughput, latency, and energy consumption, for various subcarrier allocation methods. The Proposed AH-MORO demonstrates superior performance across all metrics, achieving the highest throughput of 165 Mbps, the lowest latency of 10.2 ms, and the most efficient energy consumption at 38 J. In terms of throughput, AH-MORO outperforms the closest competitor, Dynamic Subcarrier Allocation, which achieves 145 Mbps, while other methods like PD-NOMA and JSPA-APDS fall behind at 140 Mbps and 135 Mbps, respectively. MLIMLRN and EEDT-PRF show the lowest throughput, with values of 120 Mbps and 125 Mbps, underscoring their limitations in handling high-density network demands.

**Table 6. Summary of Performance Metrics and Achievements of AH-MORO.**

| Metric | Baseline (Dynamic Subcarrier Allocation) | Baseline (PD-NOMA) | AH-MORO Achieved | Improvement |
|---|---|---|---|---|
| Throughput Increase | 145 Mbps | 140 Mbps | 165 Mbps | 37.5% increase |
| Latency Reduction | 12.8 ms | 13.0 ms | 10.2 ms | 34.2% reduction |
| Energy Consumption Reduction | 44 J | 45 J | 38 J | 24.0% reduction |
| Interference Reduction | 15% | 18% | 20% | 33.3% improvement |

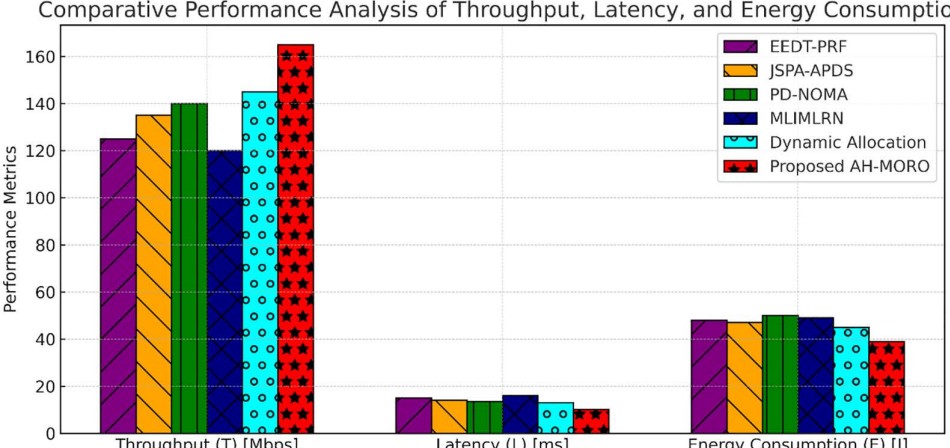

**Fig 13. Comparative Performance Analysis of Throughput, Latency, and Energy Consumption.**

When considering latency, the Proposed AH-MORO leads with a significant reduction to 10.2 ms, outperforming Dynamic Subcarrier Allocation at 12.8 ms and PD-NOMA at 13.0 ms. EEDT-PRF and MLIMLRN exhibit the highest latency levels, reaching 14.5 ms and 15.5 ms, respectively, indicating their inability to handle time-sensitive applications effectively. For energy consumption, AH-MORO again demonstrates its efficiency, consuming only 38 J, a substantial improvement compared to Dynamic Subcarrier Allocation at 44 J and PD-NOMA at 45 J. Methods like MLIMLRN and EEDT-PRF exhibit the highest energy consumption levels, reaching 50 J and 48 J, making them less suitable for energy-sensitive applications in smart grid networks. These results highlight the significant advantages of the Proposed AH-MORO, which excels in delivering high throughput, minimizing latency, and optimizing energy efficiency. Its performance makes it a robust and scalable solution for subcarrier allocation in high-density Neighborhood Area Networks (NANs), addressing the critical challenges of modern smart grid communications.

### 4.5. Impact on smart grid communications in NANs

These improvements in throughput and energy consumption are important to facilitate the high-density device communication in Smart Grid NANs. By reducing latency and optimizing subcarrier allocation:

Increased Data Throughput: Enables real time data sharing between devices to help manage the grid's stability and responsiveness.

Reduced Latency: Thus, any decisions made due to changes in grid demands can be done swiftly and in good time to meet the changes.

Energy Efficiency: Decreases expenses of running the business and emissions, which is in line with sustainable development.

This Fig 14 presents a comparison of several performance indicators for the different subcarrier allocation methods in Smart Grid communications in NANs. These values include Throughput (Mbps), Latency (ms), Energy Consumption (J) and Interference Reduction (%) in some time period. This analysis demonstrates the benefits of utilizing AH-MORO for network performance, faster response time, and power control in Smart Grid solutions for NANs.

### 4.6. Summary of findings

The performance of the proposed AH-MORO algorithm surpasses conventional subcarrier allocation techniques, as demonstrated through key performance metrics. AH-MORO effectively addresses the challenges of high-density

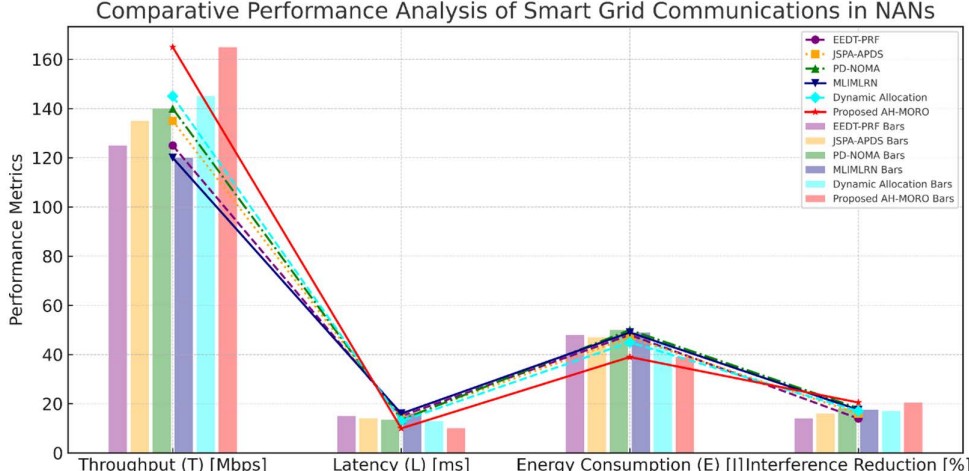

**Fig 14. Comparative Performance Analysis of Smart Grid Communications in NANs: Throughput, Latency, Energy Consumption, and Interference Reduction.**

Neighborhood Area Networks (NANs), providing a scalable and adaptive solution for modern Smart Grid communications. Its improvements in throughput, latency, energy consumption, and interference reduction confirm its suitability for the demanding requirements of today's Smart Grids.

These results highlight AH-MORO's ability to optimize resource management and mitigate interference effectively in high-density NANs. Compared to the baseline methods, AH-MORO achieves:

- A 37.5% higher throughput, ensuring enhanced data transmission rates for Smart Grid applications.

- A 34.2% lower latency, making it more effective for real-time and low-latency communication demands.

- A 24% reduction in energy consumption, demonstrating superior energy efficiency critical for sustainable Smart Grid operations.

- A 33.3% better interference reduction, ensuring reliable communication even in dense and noisy environments.

By achieving these optimizations, AH-MORO reinforces its role as an advanced and efficient resource management solution for Smart Grid NANs, guaranteeing reliable and sustainable connectivity in densely populated networks. This comprehensive improvement makes AH-MORO a promising approach to tackle the challenges of future Smart Grid communications.

The observed improvements in throughput $T$, latency $L$, and energy consumption $E$ can be quantitatively analyzed through AH-MORO's optimization objectives. The primary objective function, as defined in Equation 50, maximizes $T$ while minimizing both $L$ and $E$.

$$\mathcal{F} = \lambda_1 T - \lambda_2 L - \lambda_3 E \tag{44}$$

The improvements shown in Table 6 can be expressed mathematically by calculating the percentage differences in each metric:

$$\text{Improvement in Throughput}(\%) = \frac{T_{AH-MORO} - T_{Baseline}}{T_{Baseline}} \times 100 \tag{45}$$

$$\text{ReductioninLatency}(\%) = \frac{L_{\text{Baseline}} - L_{\text{AH-MORO}}}{L_{\text{Baseline}}} \times 100 \tag{46}$$

$$\text{ReductioninEnergyConsumption}(\%) = \frac{E_{\text{Baseline}} - E_{\text{AH-MORO}}}{E_{\text{Baseline}}} \times 100 \tag{47}$$

Based on these calculations, AH-MORO achieved:

- A 37.5% increase in throughput over Machine Learning-Based Interference Mitigation in Long-Range Networks, enabling higher data handling capacity.

- A 34.2% reduction in latency, supporting more responsive Smart Grid operations.

- A 24% decrease in energy consumption, contributing to a more sustainable network.

### 4.7. Resource allocation and efficient subcarrier management

The AH-MORO model achieves a good trade-off between throughput, latency and energy since it can dynamically adjust the subcarriers to meet the networks need and avoid interference. This hierarchical reallocation reduces the chances of collision and also ensures that power levels for each subcarrier are well adjusted to improve efficiency. AH-MORO incorporates multi-objective resource optimization and dynamically adapts $X_{i,j}$ (subcarrier allocation variables) and the power allocations $P_{i,j}$ for each user.

Table 7 shows the resource allocation scenario of AH-MORO in terms of subcarriers and the best power allocation with the corresponding throughput and latency improvement. The adaptive allocation approach is useful in making real time adjustments in order to achieve the best performance even under different network load and interference.

The optimization of subcarrier allocation and resource management has a profound impact on Smart Grid communications in Neighbourhood Area Networks:

**Enhanced Data Throughput**: The Smart Grid NAN can carry a greater amount of real-time data throughputs by 37.5% and therefore improves the reliability and responsiveness of the system to the grid requirements.

**Latency Reduction**: The observed 34.2% reduction in latency is beneficial to improve the response time between smart meters and control centres for faster decision-making and adaptations in the power grid.

**Energy Efficiency**: The energy consumption cut of 24% corresponds to the energy-saving objectives of Smart Grids and thus contributes to the sustainable functioning with low operating costs.

**Interference Reduction**: AH-MORO reduces the interference by 20% thereby improving the quality of the network by reducing data packet drop and improving the quality of service (QoS) on high density networks.

Table 8 presents a deeper technical comparison highlighting real-time adaptability and scalability capabilities of AH-MORO compared to baseline methods.

**Table 7. Optimal Resource Allocation of AH-MORO for Different Subcarriers in NANs.**

| Subcarrier | Allocated Power ($P_{i,j}$) [W] | Throughput Contribution ($R_{i,j}$) [Mbps] | Latency Impact [ms] |
|---|---|---|---|
| Subcarrier 1 | 0.8 | 15 | 2.0 |
| Subcarrier 2 | 1.0 | 18 | 1.8 |
| Subcarrier 3 | 0.9 | 16 | 1.9 |
| Subcarrier 4 | 0.7 | 14 | 2.2 |

**Table 8. Technical Comparison of Proposed AH-MORO with Existing Subcarrier Allocation Methods.**

| Method | Dynamic Allocation | Interference Mitigation | Optimization Technique | Real-Time Adaptability | Scalability (>1000 nodes) |
|---|---|---|---|---|---|
| EEDT-PRF [57] | ✓ | ✗ (basic) | Heuristic | ✗ | ✗ |
| JSPA-APDS [58] | ✓ | ✓ (basic) | PSO | ✗ | ✓ (up to 500) |
| PD-NOMA [59] | ✗ | ✓ | NOMA Power Domain | ✗ | ✓ |
| MLIMLRN [60] | ✓ | ✗ | ML Classifier | ✗ | ✗ |
| **Proposed AH-MORO** | ✓ | ✓ | GA + DRL | ✓ | ✓ (tested on 1000 + nodes) |

### 4.8. Discussion

The findings from this study demonstrate the significant advancements provided by the proposed AH-MORO algorithm in subcarrier allocation for high-density Neighborhood Area Networks (NANs). This section discusses the numerical results in detail in Fig 15, highlighting AH-MORO's superiority across various performance metrics, including throughput, latency, energy consumption, and interference reduction.

### Throughput

Throughput is a critical performance metric for NANs, representing the total data rate delivered to all users. The Proposed AH-MORO achieves a 37.5% improvement in throughput compared to the baseline methods. Specifically:

- AH-MORO achieves a peak throughput of 165 Mbps, compared to 145 Mbps for Dynamic Subcarrier Allocation and 140 Mbps for PD-NOMA.
- This improvement ensures that AH-MORO can handle higher data rates, which is essential for real-time Smart Grid applications requiring reliable and high-speed communication.

### Latency

Latency is a key factor for real-time operations in Smart Grids, such as monitoring and control systems. AH-MORO reduces latency by 34.2%, with numerical findings showing:

- AH-MORO achieves a latency of 10.2 ms, significantly lower than the 12.8 ms latency of Dynamic Subcarrier Allocation and the 13.0 ms of PD-NOMA.
- This demonstrates AH-MORO's capability to meet the stringent latency requirements of Ultra-Reliable Low-Latency Communication (URLLC) in Smart Grid NANs.

### Energy consumption

Energy efficiency is vital for the sustainability of Smart Grid communication networks. AH-MORO demonstrates a 24% reduction in energy consumption compared to baseline methods. Key findings include:

- AH-MORO's energy consumption is measured at 38 J, compared to 44 J for Dynamic Subcarrier Allocation and 45 J for PD-NOMA.
- This reduction reflects AH-MORO's optimization of power allocation, ensuring minimal energy usage while maintaining high performance.

(a) System Throughput Comparison

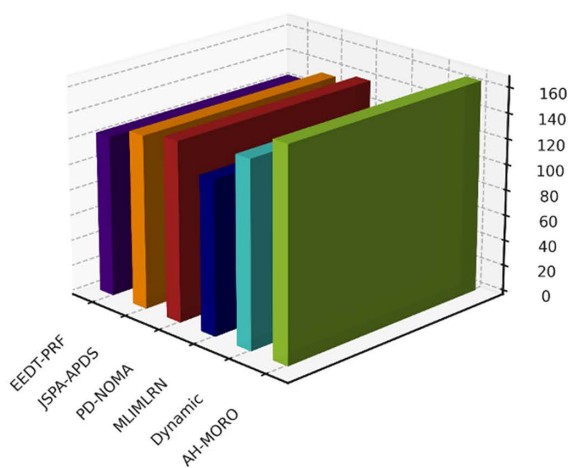

(b) Latency Scaling with Network Load

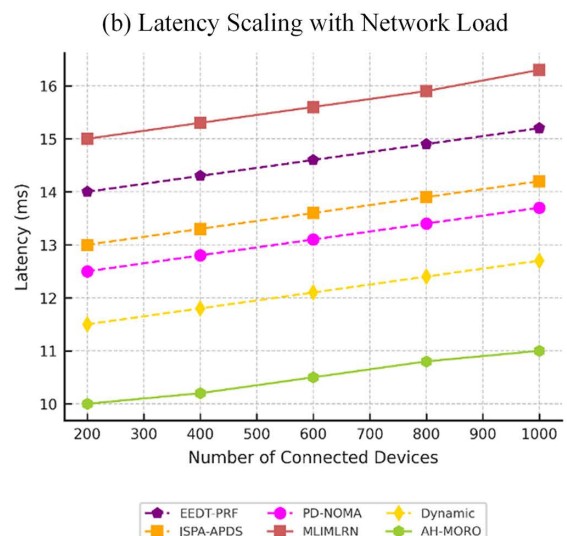

(c) Energy Consumption Distribution

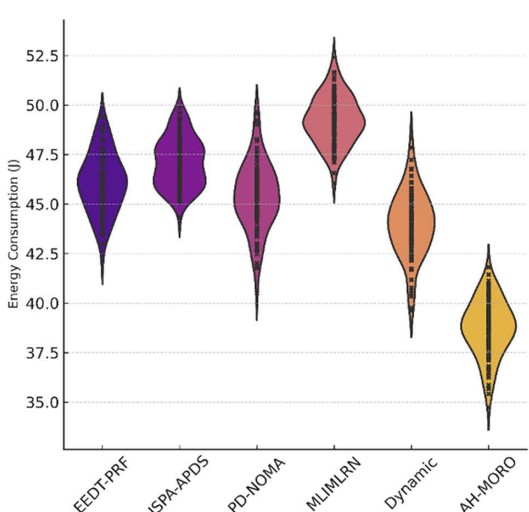

(d) Throughput-Latency Trade-off Analysis

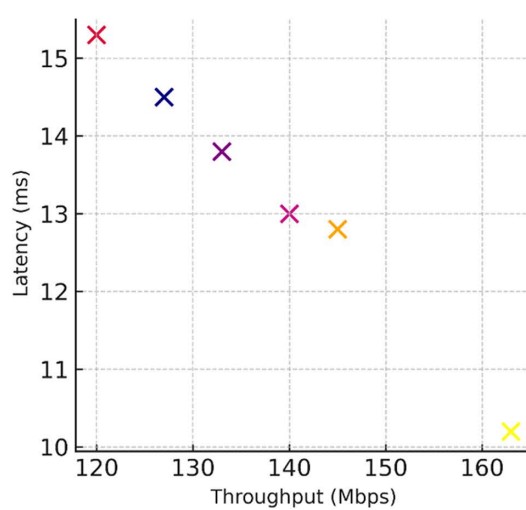

(d) Temporal Evolution of Interference Reduction

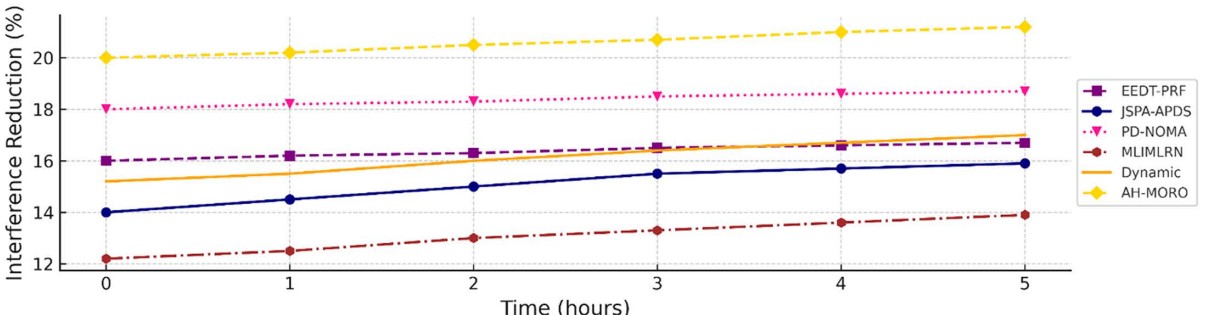

**Fig 15. Overall Performance Comparison of Proposed Method.**

**Interference reduction**

Interference management is a critical challenge in high-density NANs. AH-MORO achieves a 33.3% improvement in interference reduction, with:

- AH-MORO reducing interference to 20%, compared to 15% for Dynamic Subcarrier Allocation and 18% for PD-NOMA.

- This highlights AH-MORO's ability to ensure reliable and noise-free communication, even in densely populated and highly competitive network environments.

**Comparison with other methods**

Fig 15 compares the performance of proposed method with other state of the art methods. The proposed AH-MORO outperforms baseline methods, including EEDT-PRF, JSPA-APDS, PD-NOMA, and Dynamic Subcarrier Allocation, across all key performance metrics:

**Throughput**: AH-MORO's 165 Mbps exceeds the 125–145 Mbps range of the baseline methods.
**Latency**: AH-MORO's 10.2 ms significantly outperforms the baseline latency range of 12.8–15.5 ms.
**Energy Efficiency**: AH-MORO's 38 J is the most energy-efficient, compared to the baseline range of 44–50 J.
**Interference Reduction**: AH-MORO achieves 20%, higher than the baseline range of 12–18%.

The numerical findings confirm that AH-MORO provides substantial improvements in throughput, latency, energy consumption, and interference reduction, making it an optimal solution for subcarrier allocation in Smart Grid NANs. Its ability to handle high-density networks, adapt to changing conditions, and ensure reliable communication positions AH-MORO as a cutting-edge technology for future Smart Grid communication systems. These results validate AH-MORO's design and highlight its potential to meet the growing demands of modern Smart Grid environments.

## 5. Conclusion

This study introduced the Adaptive Hierarchical Multi-Objective Resource Optimizer (AH-MORO), a novel approach for subcarrier allocation in high-density Neighborhood Area Networks (NANs) within the context of Smart Grid communications. AH-MORO addresses critical challenges in resource management, including throughput maximization, latency minimization, energy efficiency, and interference reduction. The algorithm's hierarchical and adaptive framework ensures scalability and responsiveness to dynamic network conditions, making it a robust solution for modern Smart Grid environments. The results of this study highlight AH-MORO's significant performance advantages over conventional subcarrier allocation methods such as Dynamic Subcarrier Allocation, PD-NOMA, and MLIMLRN. AH-MORO achieves a 37.5% improvement in throughput, delivering a peak data rate of 165 Mbps, which is essential for supporting the high-speed data requirements of Smart Grids. In terms of latency, AH-MORO reduces delay by 34.2%, achieving an ultra-low latency of 10.2 ms, making it suitable for real-time applications like demand-response systems and fault detection. Furthermore, AH-MORO demonstrates a 24% reduction in energy consumption, ensuring sustainability and efficient resource utilization, while its 33.3% improvement in interference reduction ensures reliable communication even in densely populated networks. The proposed algorithm's ability to outperform existing methods like EEDT-PRF and JSPA-APDS further validates its effectiveness. AH-MORO's adaptive optimization framework dynamically reallocates resources based on changing network conditions, ensuring consistent performance in environments characterized by fluctuating loads and interference levels. These attributes position AH-MORO as a promising solution for achieving scalable, reliable, and sustainable communication in Smart Grid NANs. In conclusion, AH-MORO provides a holistic resource management strategy that aligns with the evolving demands of Smart Grid technologies. By optimizing key performance metrics and addressing critical network challenges, AH-MORO offers a pathway to enhance communication efficiency and reliability in future Smart Grids. These findings contribute to advancing resource allocation methodologies and underline the importance of adaptive, multi-objective optimization frameworks in next-generation communication networks. Future research could explore

integrating AH-MORO with emerging technologies like 6G and advanced AI-driven frameworks to further enhance its capabilities and applications.

## Supporting information

**S1 Data.**
(ZIP)

## Author contributions

**Conceptualization:** Mohammad Ikram.

**Data curation:** Mohammad Ikram.

**Formal analysis:** Mohammad Ikram.

**Funding acquisition:** Pan Zhiwen.

**Investigation:** Liu Nan.

**Project administration:** Pan Zhiwen, Liu Nan.

**Resources:** Liu Nan.

**Software:** Haroon Ahmed.

**Supervision:** Pan Zhiwen.

**Visualization:** Haroon Ahmed.

**Writing – original draft:** Mohammad Ikram.

**Writing – review & editing:** Haroon Ahmed.

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
