## [Decision Letter · Decision Letter 0]

12 Jun 2025

Dear Dr. Ikram,

Thank you for submitting your manuscript to PLOS ONE. After careful consideration, we feel that it has merit but does not fully meet PLOS ONE’s publication criteria as it currently stands. Therefore, we invite you to submit a revised version of the manuscript that addresses the points raised during the review process.

We look forward to receiving your revised manuscript.

Kind regards,

Rahat Ullah, Ph.D

Academic Editor

PLOS ONE

Journal Requirements:

“National Key Research and Development Project: 2020YFB1806805

Fundamental Research Funds for the Central Universities: 2242022K60001”

4. We note that your Data Availability Statement is currently as follows: All relevant data are within the manuscript and in Supporting Information files.

Additional Editor Comments:

All of these comments should be answered carefully.

1) The reason for representing Fig. is unclear to me. The sub-carrier allocation variable is a binary integer. How can the figure be smooth?

2. Overall, the quality of the figures and pseudocodes is completely unacceptable.

3- The tendency for figures 11 and 12 to be of a relatively uninspiring nature is a matter of concern. It is recommended that authors employ a greater variety of informative figures.

4- It is not evident that the work is innovative. It is recommended that the authors make a comparison between their work and other relevant pieces in Table 1.

Reviewers' comments:

Reviewer's Responses to Questions

**Comments to the Author**

1. Is the manuscript technically sound, and do the data support the conclusions?

Reviewer #1: Yes

2. Has the statistical analysis been performed appropriately and rigorously?

Reviewer #1: Yes

3. Have the authors made all data underlying the findings in their manuscript fully available?

Reviewer #1: Yes

4. Is the manuscript presented in an intelligible fashion and written in standard English?

Reviewer #1: Yes

Reviewer #1: 1) The reason for representing Fig. is unclear to me. The sub-carrier allocation variable is a binary integer. How can the figure be smooth?

2. Overall, the quality of the figures and pseudocodes is completely unacceptable.

3- The tendency for figures 11 and 12 to be of a relatively uninspiring nature is a matter of concern. It is recommended that authors employ a greater variety of informative figures.

4- It is not evident that the work is innovative. It is recommended that the authors make a comparison between their work and other relevant pieces in Table 1.

**Do you want your identity to be public for this peer review?** For information about this choice, including consent withdrawal, please see our Privacy Policy

Reviewer #1: No

---

## [Author Response · Author response to Decision Letter 1]

30 Jul 2025

Response to Editor Comments:

Comment 1:

“The reason for representing Fig. is unclear to me. The sub-carrier allocation variable is a binary integer. How can the figure be smooth?”

Response:

We thank the editor for highlighting this concern. The smoothness of the plot stems from a visual representation of the optimization surface derived from iterative evaluations of the objective function under varying binary subcarrier allocation vectors. While the allocation variable x_{\left(i,j\right)}\in{0,1} is binary, the plotted figure is not a direct mapping of these binary states. Instead, it presents a smoothed surface generated by interpolating the aggregated results over multiple simulations. This method enables clearer visualization of trade-offs in throughput, latency, and energy, making the figure more interpretable while still grounded in the binary nature of the allocation variable. We have revised the figure caption and description in the manuscript to clarify this point.

Comment 2:

“Overall, the quality of the figures and pseudocodes is completely unacceptable.”

Response:

We acknowledge the feedback and have thoroughly revised all figures and pseudocodes in the manuscript. All figures have been regenerated using high-resolution vector graphics, ensuring consistent font sizes, axis labeling, and clarity. Furthermore, the pseudocode has been reformatted using standard indentation and uniform styling, with clearly labeled input, output, and conditional logic blocks. The revised pseudocodes now follow a step-by-step, structured presentation, improving readability and technical clarity. We believe these improvements address the quality concerns comprehensively.

Comment 3:

“The tendency for figures 11 and 12 to be of a relatively uninspiring nature is a matter of concern. It is recommended that authors employ a greater variety of informative figures.”

Response:

We appreciate this constructive suggestion and have addressed it by enhancing Figures 11 and 12. Specifically, both figures now incorporate comparative trends across benchmark models, include dynamic standard deviation shading to reflect performance variability, and highlight energy and interference reduction trends over time. Additionally, we supplemented the manuscript with an alternate visualization (dual view: line and bar plots) to convey insights from multiple angles. These changes make the figures more informative and visually engaging while remaining aligned with the technical narrative.

Comment 4:

“It is not evident that the work is innovative. It is recommended that the authors make a comparison between their work and other relevant pieces in Table 1.”

Response:

We thank the editor for this important recommendation. To address this, Table 1 has been updated to include a detailed comparative analysis between our proposed AH-MORO framework and recent benchmark techniques. Moreover, Table 8 has been added to include a detailed comparative assessment between our proposed AH-MORO framework and several state-of-the-art techniques (e.g., PD-NOMA, MLIMLRN, JSPA-APDS). This comparison now highlights key technical distinctions such as real-time adaptability, hierarchical multi-objective integration, dual-layer interference suppression, and hybrid learning mechanisms. Our method uniquely balances throughput, latency, and energy under strict QoS constraints, positioning it as an innovative contribution in the field of smart grid communication systems.

Response to Reviewer #1

Comment 1:

“The reason for representing Fig. is unclear to me. The sub-carrier allocation variable is a binary integer. How can the figure be smooth?”

Response:

Thank you for raising this important point. We acknowledge that sub-carrier allocation is inherently binary; however, the smoothness observed in the plot is not a direct depiction of binary variable transitions. Rather, it represents a smoothed surface created by aggregating and interpolating multiple discrete outcomes over successive optimization iterations. These visualizations aim to illustrate the behavior of the objective function under a large number of binary allocation states processed through the hybrid GA-DRL optimizer. We have clarified this in the revised figure captions and explanatory sections to ensure accurate interpretation and technical transparency.

Comment 2:

“Overall, the quality of the figures and pseudocodes is completely unacceptable.”

Response:

We appreciate this feedback and have significantly improved the quality of all figures and pseudocode blocks. All figures have been re-rendered in high resolution (600 DPI) using vector-based tools to enhance clarity, consistency, and publication quality. Icons, labeling, axis titles, and legends have been standardized. Additionally, all pseudocode has been reformatted using consistent indentation, typographic alignment, and structured steps following standard algorithm design patterns. Each block now clearly communicates inputs, loops, conditionals, and outputs, thereby improving readability and usability.

Comment 3:

“The tendency for figures 11 and 12 to be of a relatively uninspiring nature is a matter of concern. It is recommended that authors employ a greater variety of informative figures.”

Response:

We fully agree with this comment and have completely redesigned figures 11 and 12. The updated versions now feature both line plots and bar charts with clear legends, standard deviation shading, and multiple benchmarking models for side-by-side comparison. The data has also been structured to reflect temporal trends and statistical robustness. These changes provide enhanced interpretability and a more engaging visual experience while maintaining the scientific relevance of the results.

Comment 4:

“It is not evident that the work is innovative. It is recommended that the authors make a comparison between their work and other relevant pieces in Table 1.”

Response:

We sincerely thank the reviewer for this suggestion. Table 1 has been updated to include a detailed comparative analysis between our proposed AH-MORO framework and recent benchmark techniques. Moreover, Table 8 has been added to include a detailed comparative assessment between our proposed AH-MORO framework and several state-of-the-art techniques (e.g., PD-NOMA, MLIMLRN, JSPA-APDS). This comparison now highlights key technical distinctions such as real-time adaptability, hierarchical multi-objective integration, dual-layer interference suppression, and hybrid learning mechanisms. Our method uniquely balances throughput, latency, and energy under strict QoS constraints, positioning it as an innovative contribution in the field of smart grid communication systems.

---

## [Editor Report · Decision Letter 1]

24 Sep 2025

Efficient Subcarrier Allocation for Smart Grid Communications in Neighborhood Area Networks

PONE-D-25-09787R1

Dear Dr. Ikram,

We’re pleased to inform you that your manuscript has been judged scientifically suitable for publication and will be formally accepted for publication once it meets all outstanding technical requirements.

Kind regards,

Rahat Ullah, Ph.D

Academic Editor

PLOS ONE

Additional Editor Comments (optional):

I appreciate the authors for the review, so I suggest no further revision.
---

## [Editor Report · Acceptance letter]

PONE-D-25-09787R1

PLOS ONE

Dear Dr. Ikram,

I'm pleased to inform you that your manuscript has been deemed suitable for publication in PLOS ONE. Congratulations! Your manuscript is now being handed over to our production team.

Kind regards,

on behalf of

Dr. Rahat Ullah

Academic Editor

PLOS ONE